# LOCAL MAP SAMPLING FOR DIFFUSION MODELS

## ABSTRACT

Diffusion Posterior Sampling (DPS) provides a principled Bayesian approach to inverse problems by sampling from $p(x_0 \mid y)$. While posterior sampling is valuable for capturing uncertainty and multi-modality, many classical and practical inverse problem settings ultimately prioritize accurate point estimation—most notably the MAP estimator, which has long served as a standard reconstruction objective in imaging and scientific applications. We introduce *Local MAP Sampling (LMAPS)*, a new inference framework that iteratively solving local MAP subproblems along the diffusion trajectory. This perspective clarifies their connection to global MAP and DPS, offering a unified probabilistic interpretation for optimization-based methods. Building on this foundation, we develop practical algorithms with a covariance approximation motivated by Gaussian prior assumption, a reformulated objective for stability and interpretability. Across a broad set of image restoration and scientific tasks, LMAPS achieves the state-of-the-art performance.

## 1 INTRODUCTION

Diffusion Posterior Sampling (DPS) is a recently proposed framework that extends diffusion generative models to Bayesian inference (Chung et al., 2022; Song et al., 2023c). This framework is particularly powerful for a wide range of applications, ranging from combined guidance and style transfer (Ye et al., 2024) to inverse problems such as medical imaging (Chung & Ye, 2022), image restoration (Chung et al., 2022), and scientific data reconstruction (Zheng et al., 2025), where it enables high-quality reconstructions while also providing principled uncertainty quantification (Ye et al., 2024). DPS conditions the generative process on observed measurements, enabling efficient sampling from posterior distributions over clean data $p(x_0 \mid y)$. This group of approaches and variants includes but not limited to TMPD (Boys et al., 2023), DDNM (Wang et al., 2022), ΠGDM (Song et al., 2023b), TFG (Guo et al., 2025).

While posterior sampling is fundamentally important in Bayesian inverse problems—capturing multi-modality, providing calibrated uncertainty, and supporting downstream decision making through credible intervals and risk-sensitive criteria—there is a parallel and long-standing line of work that emphasizes point estimation, and in particular MAP, as an equally central objective. Classical treatments of Bayesian inverse problems show that the MAP estimator often coincides with the solution of a variationally regularized optimization problem and is widely used as a practical reconstruction rule in imaging, medical, and geophysical applications (Stuart, 2010; Kaipio & Somersalo, 2005; Tarantola, 2005).

Optimization-based approaches—such as Resample (Song et al., 2023a), DiffPIR (Zhu et al., 2023), DCDP (Li et al., 2024), and DMPlug (Wang et al., 2024)—have shown strong performance by alternating between denoising, optimization, and resampling to address inverse problems. Unlike DPS, which attempts to sample from the posterior distribution $p(x_0 \mid y)$, optimization-based approaches prioritize reconstruction performance over distributional faithfulness. Nevertheless, it's still unclear if the iterative procedure converges to the global MAP solution, i.e., $\arg \max p(x_0 \mid y)$, would it still be consistent with DPS? Clarifying this foundation could provide both a principled interpretation and a stronger theoretical basis for optimization-based methods.

In this work, we argue that the optimization steps in these methods inherently solve a *local MAP problem*. But the resulting solutions neither converge to the global MAP nor equivalent to posterior sampling. Instead, they are more likely to reflect a trade-off between the two.

(a) Relationship between LMAPS and other methods.

(b) Unconditional generation process.

(c) LMAPS generation process.

Figure 1: Comparison of LMAPS with other methods. (a). The relationship between different alignment approaches; (b). The generation process of unconditional diffusion model; (c). The generation process of LMAPS.

Our main contributions are summarized as follows:

- **Theoretical.** We formulate *Local MAP Sampling (LMAPS)*, a new inference framework that iteratively solves local maximum-a-posteriori subproblems along the diffusion trajectory. We analyze its relationship to global MAP and DPS, and show that LMAPS unifies Tweedie Moment Projected Diffusion (TMPD) and optimization-based inverse problem methods under a single framework. The relationship between LMAPS and existing methods are presented in Figure 1.

- **Methodological.** To address inverse problems, we introduce a covariance approximation motivated by Gaussian prior assumption. In addition, we propose an objective reformulation that improves interpretability and enhances numerical stability.

- **Empirical.** LMAPS is validated on 10 image restoration tasks (linear, nonlinear, non-differentiable) and 3 scientific inverse problems. It achieves the best results in $43/60$ FFHQ/ImageNet cases, while being more efficient than DAPS. On scientific tasks, LMAPS consistently attains the highest PSNR, including $> 1.5$ dB gains on 3 linear inverse scattering tasks.

## 2 BACKGROUND

**Unconditional diffusion models**. The goal of diffusion model is to sample from an unknown distribution $\pi_0(x_0)$ given a training dataset $\mathcal{D} = \{x_0^i\}_{i=1}^N$. Given a data point $x_0 \sim \pi_0$ and a time step $t$, a noisy datapoint is sampled from the transition kernel: $p_t(x_t \mid x_0) = \mathcal{N}(x_t; \alpha_t x_0, \sigma_t^2 \mathbb{I})$. Diffusion process is built by mixture of densities: $p_t(x_t) = \int p_t(x_t \mid x_0)\pi_0(x_0)dx_0$, and DDIM samples $\pi_0(x_0)$ by running an iterative process $p_t(x_t)$ from time $t = T$ to $t = 0$ with the initial condition $x_T \sim p(x_T)$:

$$x_{t-\Delta t} = g(m_{0|t}(x_t), x_t, \epsilon), \quad \epsilon \sim \mathcal{N}(0, \mathbb{I}) \tag{1}$$

where $\epsilon \sim \mathcal{N}(0, \mathbb{I})$ is the fresh noise added at the inference time, $m_{0|t}(t, x) = \mathbb{E}[x_0 \mid x_t]$ is the ideal denoiser, and we define:

$$g(\xi, x_t, \epsilon) := \alpha_{t-\Delta t}\xi + \sigma_{t-\Delta t}\left(\sqrt{1 - \rho_t^2}\frac{x_t - \alpha_t\xi}{\sigma_t} + \rho_t\epsilon\right), \tag{2}$$

The goal of posterior sampling is to generate samples under some condition $y$, i.e., sample $x_0$ from a posterior distribution, $\pi_{0|y}(x_0 \mid y)$, where $y$ could be class labels, measurements or text information, for example. In this paper, we focus on two representative lines of posterior sampling approaches with diffusion priors: (i) the family of diffusion posterior sampling (DPS) methods based on Tweedie's formula, and (ii) Decoupled Annealing Posterior Sampling (DAPS).

**Diffusion Posterior Sampling (DPS) family**. DPS generate $x_0 \sim \pi_{0|y}(x_0 \mid y)$ by running an iterative process $p_{t|y}(x_t \mid y)$ from time $t = T$ to $t = 0$ with the initial condition $x_T \sim p(x_T \mid y)$:

$$x_{t-\Delta t} = g(m_{0|t,y}(t, x_t, y), x_t, \epsilon), \quad \epsilon \sim \mathcal{N}(0, \mathbb{I}), \tag{3}$$

| **Algorithm 1** DPS | **Algorithm 2** DAPS | **Algorithm 3** LMAPS |
|---|---|---|
| 1: **Input:** $x_{t_N} \sim \pi_T$ | 1: **Input:** $x_{t_N} \sim \pi_T$ | 1: **Input:** $x_{t_N} \sim \pi_T$ |
| 2: **for** $k = N$ to 1 **do** | 2: **for** $k = N$ to 1 **do** | 2: **for** $k = N$ to 1 **do** |
| 3: $\quad \tilde{x}_0 = \mathbb{E}[x_0 \mid x_{t_k}, y]$ | 3: $\quad \tilde{x}_0 \sim p(x_0 \mid x_{t_k}, y)$ | 3: $\quad \tilde{x}_0 = \arg\max p(x_0 \mid x_{t_k}, y)$ |
| 4: $\quad \epsilon \sim \mathcal{N}(0, \mathbb{I})$ | 4: $\quad \epsilon \sim \mathcal{N}(0, \mathbb{I})$ | 4: $\quad \epsilon \sim \mathcal{N}(0, \mathbb{I})$ |
| 5: $\quad x_{t_{k-1}} = g(\tilde{x}_0, x_{t_k}, \epsilon)$ | 5: $\quad x_{t_{k-1}} \sim \mathcal{N}(\alpha_{t_{k-1}} x_0, \sigma_{t_{k-1}}^2 \mathbb{I})$ | 5: $\quad x_{t_{k-1}} = g(\tilde{x}_0, x_{t_k}, \epsilon)$ |
| 6: **end for** | 6: **end for** | 6: **end for** |
| 7: **return** $x_0$ | 7: **return** $x_0$ | 7: **return** $x_0$ |

Figure 2: Comparison of inference algorithm between DPS, DAPS and LMAPS.

where $m_{0|t,y}(t, x_t, y) = \mathbb{E}[x_0 \mid x_t, y]$ is the conditional denoiser. According Tweedie's formula,

$$\mathbb{E}[x_0 \mid x_t, y] = m_{0|t} + \frac{\sigma_t^2}{\alpha_t} \nabla_{x_t} \log p(y \mid x_t). \tag{4}$$

Eq. (4) connects the conditional denoiser $\mathbb{E}[x_0 \mid x_t, y]$ with the unconditional denoiser $\mathbb{E}[x_0 \mid x_t]$. However, the additional term $\nabla_{x_t} \log p(y \mid x_t)$ is still intractable. One can train a neural network to approximate $\nabla_{x_t} \log p(y \mid x_t)$, like classifier guidance (Dhariwal & Nichol, 2021). Training-free guidance, such as in (Chung et al., 2022), usually approximates $\nabla_{x_t} \log p(y \mid x_t)$ by a convenient single-sample approximation, $p(y \mid x_t) \approx p(y \mid m_{0|t}(x_t))$, according to chain rule:

$$\nabla_{x_t} \log p(y \mid x_t) \approx \nabla_{x_t} m_{0|t}(t, x_t) \nabla_{m_{0|t}} \log p(y \mid m_{0|t}(t, x_t)). \tag{5}$$

**Decoupled Annealing Posterior Sampling (DAPS)** (Zhang et al., 2025a). Alternatively, DAPS developed a new framework to sample $x_0 \sim \pi_{0|y}(x_0 \mid y)$, which is given by the following iterations:

$$x_{0|t,y} \sim p(x_0 \mid x_t, y)$$
$$x_{t-\Delta t} \sim \mathcal{N}(\alpha_{t-\Delta t} x_0, \sigma_{t-\Delta t}^2 \mathbb{I}). \tag{6}$$

Approximate posterior samples $x_{0|t,y}$ are obtained at each diffusion step using Langevin dynamics.

## 3 LOCAL MAP SAMPLING

### 3.1 LOCAL MAP AND GLOBAL MAP

**Global MAP.** In Bayesian inference, the maximum a posteriori (MAP) estimate is defined as the single configuration that maximizes the posterior probability,

$$x_0^{\text{MAP}} := \arg\max_{x_0} p(x_0 \mid y). \tag{7}$$

We refer to this as the *global MAP*, since it directly targets the mode of the full posterior distribution after conditioning on the observation $y$. Unlike posterior sampling methods (e.g., DPS or DAPS), which produce diverse draws from $p(x_0 \mid y)$, global MAP yields a point estimate corresponding to (one of) the maximizers of the posterior. This estimate prioritizes fidelity and certainty over diversity, offering a principled way to recover a solution that best aligns with both the diffusion prior and the measurement model.

**Local MAP.** Directly solving for $x_0^{\text{MAP}}$ in high-dimensional, non-convex posteriors can be computationally intractable. Instead, we consider a sequence of *local MAP* problems, which implemented by DDIM-like iteration from time $t = T$ to $t = 0$ with the initial condition $x_T \sim p(x_T \mid y)$:

$$x_0^*(t, x_t, y) := \arg\max p(x_0 \mid x_t, y), \tag{8a}$$
$$x_{t-\Delta t} = g(x_0^*, x_t, \epsilon), \quad \epsilon \sim \mathcal{N}(0, \mathbb{I}). \tag{8b}$$

Eq. (8a) and Eq. (8b) correspond to the local MAP step and the DDIM update step, respectively. In particular, the local MAP step is equivalent to:

$$x_0^*(t, x_t, y) = \arg\min\{-\log p(x_0 \mid x_t) - \log p(y \mid x_0)\}. \tag{9}$$

This optimization problem can be solved via gradient descent if $\log p(x_0 \mid x_t)$ and $\log p(y \mid x_0)$ are known and differentiable, although in practice we approximate $p(x_0 \mid x_t)$ as discussed in Sec. 4.

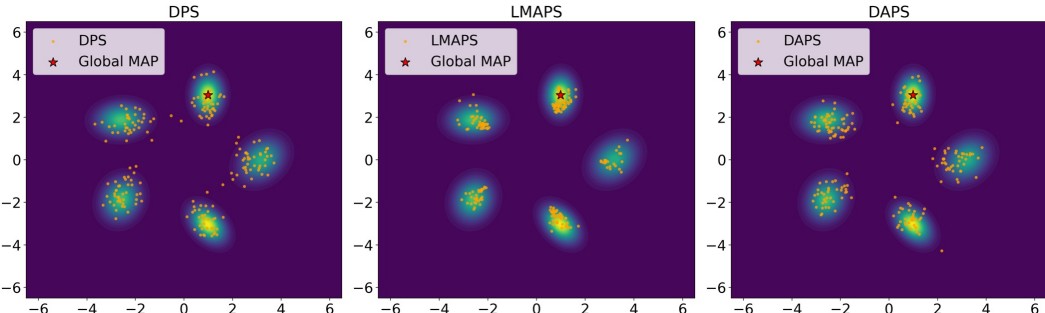

Figure 3: Comparison of LMAPS, DPS, DAPS and Global MAP on 2D synthetic data, here we assume $p(x_0 \mid y)$ is a Gaussian mixture which have analytical expression (see App. A). LMAPS is less likely to generate samples in the between-mode regions or low-density regions.

## 3.2 THE DIFFERENCE BETWEEN DPS, LOCAL MAP AND GLOBAL MAP

One might expect that the iteration in Eq. (8) can be used to sample from the posterior $p(x_0 \mid y)$ or converge to global MAP $\arg\max p(x_0 \mid y)$. Unfortunately, this is generally not the case.

**DPS vs. local MAP.** DPS evolves $x_t$ by using the conditional mean $m_{0|t,y}(t, x_t, y) = \mathbb{E}[x_0 \mid x_t, y]$ inside the DDIM update (Eq. (3)), whereas local MAP replaces the mean with the conditional mode: $x_0^*(t, x_t, y) = \arg\max p(x_0 \mid x_t, y)$, and then plugs $x_0^*$ into the same $g(\cdot)$ transition (Eq. (8)). Consequently, replacing $\mathbb{E}[x_0 \mid x_t, y]$ with $\arg\max p(x_0 \mid x_t, y)$ alters the forward operator acting on $p_{t|y}(x_t)$ and does not preserve the posterior marginals $p_{t|y}$.

**When are DPS and local MAP equivalent?** These two coincide if and only if $\mathbb{E}[x_0 \mid x_t, y] = \arg\max p(x_0 \mid x_t, y)$, for example if $p(x_0 \mid x_t, y)$ is (uni-variate or multi-variate) Gaussian. The condition holds, e.g., in linear-Gaussian inverse problems with a Gaussian diffusion prior approximation (quadratic negative log-density), with detailed discussion in Sec. 4. Outside of this setting (nonlinear forward models, heavy-tailed likelihoods, mixture-like priors), the posterior $p(x_0 \mid x_t, y)$ is non-Gaussian and the two updates generally differ. With non-Gaussian $p(x_0 \mid x_t, y)$, local MAP introduces a mode-seeking bias and does not reproduce posterior sampling.

**Local MAP vs. global MAP.** a global MAP solution is any maximizer of $x_0^{\mathrm{MAP}} = \arg\max p(x_0 \mid y)$. Local MAP instead solves, at each time $t$, a conditioned optimization (Eq. (9)): $x_0^*(t, x_t, y) = \arg\max p(x_0 \mid x_t, y)$. Because $x_t$ itself depends on the entire past trajectory (initialization, noise schedule, and random seeds), the sequence of local maximizers need not approach the global maximizer of $p(x_0 \mid y)$ as $t \downarrow 0$.

In summary, DPS targets $p(x_0 \mid y)$, and LMAPS targets $\arg\max p(x_0 \mid x_t, y)$ at each step. Local MAP equals DPS only in Gaussian conditional settings; outside them, local MAP generally does not sample the posterior and can fail to reach the global MAP. We visualize a toy example in Figure 3. Compared to DPS and DAPS, LMAPS is less likely to generate samples in between-mode regions or low-density regions.

## 4 LOCAL MAP SAMPLING FOR INVERSE PROBLEM

The primary goal of solving an inverse problem is to recover an unknown image or signal $x_0 \in \mathbb{R}^n$ from a prior distribution, $\pi(x_0)$, and noisy measurement $y \in \mathbb{R}^m$. Mathematically, the unknown signal and the measurements are related by a forward model:

$$y = \mathcal{H}(x_0) + z \tag{10}$$

where $\mathcal{H}(\cdot) : \mathbb{R}^n \to \mathbb{R}^m$ (with $m < n$) represents the linear or non-linear forward operator, $z \in \mathbb{R}^m$ denotes the noise in the measurement domain. We assume the added noise $z$ is sampled from a Gaussian distribution $\mathcal{N}(0, \sigma_y^2 \mathbb{I})$, where $\sigma_y > 0$ denotes the noise level. The forward operator and Eq. (10) define the likelihood $p(y \mid x_0)$ for both the global or local MAP problems in Sec. 3.1.

The final ingredient for constructing a local posterior and solving the resulting MAP problem is the choice of prior $p(x_0 \mid x_t)$. While the true transition kernel of a diffusion model prior requires simulation, we can proceed as in previous work (Boys et al., 2023; Song et al., 2023b) by projecting

onto the first two moments using a Gaussian approximation, $p(x_0 \mid x_t) \approx \mathcal{N}(x_0; m_{0|t}, \Sigma_{0|t})$, where $m_{0|t}(x_t) := \mathbb{E}[x_0 \mid x_t]$. While Boys et al. (2023) show that $\Sigma_{0|t}^{\text{TMPD}}(x_t) := \mathbb{E}[(x_0 - m_{0|t})(x_0 - m_{0|t})^T \mid x_t] = \frac{\sigma_t^2}{\alpha_t} \nabla_{x_t} m_{0|t}$, we will consider flexible choices of $\Sigma_{0|t}$. Finally, the local MAP problem amounts to solving

$$x_0^* = \arg\min (x_0 - m_{0|t})^T \Sigma_{0|t}^{-1}(x_0 - m_{0|t}) + \frac{1}{\sigma_y^2}\|y - \mathcal{H}(x_0)\|^2. \tag{11}$$

We will develop methodology for approximately solving the local MAP problem for general nonlinear inverse problems in Sec. 4.1, before discussing the case of linear inverse problems in Sec. 4.2.

### 4.1 Approximated solution for nonlinear inverse problems

**Isotropic approximation of** $\Sigma_{0|t}$. For nonlinear $\mathcal{H}(\cdot)$, there is no explicit solution for $x_0^*$ and it would be more expensive to adopt the moment projection covariance $\Sigma_{0|t} = \nabla_{0|t}^{\text{TMPD}} = \frac{\sigma_t^2}{\alpha_t}\nabla_{x_t} m_{0|t}$.

For a Gaussian prior $x_0 \sim \mathcal{N}(\mu_0, \Sigma_0)$, the exact posterior covariance under the forward noising process $x_t = \alpha_t x_0 + \sigma_t \epsilon$, $\epsilon \sim \mathcal{N}(0, \mathbb{I})$ is

$$\Sigma_{0|t} = \left(\Sigma_0^{-1} + \frac{\alpha_t^2}{\sigma_t^2}\mathbb{I}\right)^{-1} = \frac{\sigma_t^2}{\alpha_t^2}\mathbb{I} + \mathcal{O}\left(\left(\frac{\sigma_t^2}{\alpha_t^2}\right)^2\right) \preceq \frac{\sigma_t^2}{\alpha_t^2}\mathbb{I}, \tag{12}$$

so the leading term is isotropic and all anisotropy appears only as higher–order corrections as $t \to 0$ (i.e., $\sigma_t^2 \to 0$ and $\alpha_t \to 1$). More generally, even for non-Gaussian priors $p(x_0)$ with a smooth log-density, the Hessian satisfies

$$\nabla_{x_0}^2\left[-\log p(x_0 \mid x_t)\right] = \frac{\alpha_t^2}{\sigma_t^2}\mathbb{I} + \nabla_{x_0}^2\left[-\log p(x_0)\right]. \tag{13}$$

As $\sigma_t^2 \to 0$, the isotropic data term $\frac{\alpha_t^2}{\sigma_t^2}\mathbb{I}$ dominates the prior curvature, implying that the local Gaussian approximation to $p(x_0 \mid x_t)$ is asymptotically isotropic. We provide a formal statement and proof in Appendix B. Motivated by the above analysis, we approximate the conditional covariance by an isotropic form $\Sigma_{0|t} \approx \frac{1}{\text{SNR}}\mathbb{I}$, where $\text{SNR} := \alpha_t^2/\sigma_t^2$. This approximation captures the leading-order behavior of the true posterior covariance as $t \to 0$. In practice, we further introduce a tunable parameter $k$ that adjusts the relative influence between the denoising estimate $m_{0|t}$ and the measurement $y$. With this modification, the MAP objective becomes

$$x_0^* = \arg\min_{x_0} \left\{ \frac{\text{SNR}}{k}\|x_0 - m_{0|t}\|^2 + \frac{1}{\sigma_y^2}\|y - \mathcal{H}(x_0)\|^2 \right\}. \tag{14}$$

**Objective Reformulation**. In the implementation, the weighting of the two terms in Eq. (14) depends on raw signal-to-noise ratios, which can vary drastically with $t$, which makes it difficult to choose the appropriate learning rate. For analysis and implementation it is convenient to reformulate Eq. (14) in a scale-invariant way. Multiplying the objective by a positive constant (which does not change the minimizer) and introducing parameters $k_1, k_2 > 0$ such that $2k_2/k_1^2 = k/(\alpha_t^2\sigma_y^2)$, we obtain the equivalent problem

$$x_0^* = \arg\min \left\{ \left(1 - \frac{\sigma_t^2}{\sigma_t^2 + k_1^2}\right)\frac{1}{2}\|x_0 - m_{0|t}\|^2 + \frac{\sigma_t^2}{\sigma_t^2 + k_1^2}k_2\|y - \mathcal{H}(x_0)\|^2 \right\}. \tag{15}$$

This reformulation has several advantages:

- **Convex-combination interpretation**. The weights can be written as $(1 - \mu_t)$ and $\mu_t$ with $\mu_t = \sigma_t^2/(\sigma_t^2 + k_1^2) \in (0, 1)$. Thus the cost is a convex combination of the prior and data fidelity terms.

- **Automatic annealing**. As $\sigma_t^2$ decreases over time, $\mu_t$ gradually shifts the objective from measurement-driven $\mu_t \approx 1$ to prior-driven ($\mu_t \approx 0$).

---

**Algorithm 4** Local MAP Sampling (LMAPS) for inverse problems.

---

1: **Input:** measurement $y$, forward operator $\mathcal{H}(\cdot)$, pretrained DM $\epsilon_\theta(\cdot)$, number of diffusion step $N$, diffusion schedule $\alpha_t$ and $\sigma_t$, number of gradient updates $K$, objective parameters $k_1$, $k_2$, learning rate $\eta$.
2: **Initialization**: $x_N \sim \mathcal{N}(0, \mathbb{I})$
3: **for** $n = N$ to 1 **do**
4:    $\hat{x}_0 \leftarrow [x_n - \sigma_n \epsilon_\theta(x_n, n)]/\alpha_n$                             ▷ Obtain predicted data
5:    $r \leftarrow \sigma_n^2/(\sigma_n^2 + k_1^2 + 10^{-6})$
6:    **Initialization**: $x_0' \leftarrow \hat{x}_0$
7:    **for** $k = K$ to 1 **do**
8:       $grad \leftarrow (x_0' - \hat{x}_0)(1 - r) + rk_2 \nabla_{x_0'} \|y - \mathcal{H}(x_0')\|^2$      ▷ Calculate gradient in Eq. (15)
9:       $x_0' = x_0' - \eta \cdot grad$
10:   **end for**
11:   $x_{n-1} \sim \mathcal{N}(\alpha_{n-1} x_0', \sigma_{n-1}\mathbb{I})$                         ▷ Forward diffusion step
12: **end for**
13: **Output** $x_0$

---

- **Interpretable parameters**. The scale $k_1$ plays the role of a trust-region parameter balancing prior and measurement, while $k_2$ is a scale factor for the consistency loss to the measurement.

- **Numerical stability**. Keep weights in $[0, 1]$ avoids extreme scaling from SNR values, improving conditioning and optimizer robustness.

In the implementation, we adopt gradient descent to solve $x_0^*$ in Eq. (15), the algorithm of LMAPS for inverse problems is provided in Algorithm 4.

**Relationship to optimization-based methods**. Previous optimization-based approaches (Song et al., 2023a; Li et al., 2024; Zhu et al., 2023) solve for $x_0^*$ through the following objective:

$$x_0^* = \arg\min \|x_0 - m_{0|t}\|^2 + \lambda_t \|y - \mathcal{H}(x_0)\|^2, \tag{16}$$

where $\lambda_t$ is a hyperparameter, often chosen heuristically without a principled basis. These methods can be viewed as special cases of our framework by setting $\Sigma_{0|t} = \lambda_t \sigma_y^2 \mathbb{I}$ in Eq. (11).

While the objectives in Eq. (16) and Eq. (15) are indeed equivalent, we found that empirical performance strongly depends on our objective reformulation and choices of weighting terms as motivated above. Further, our local MAP interpretation provides a probabilistic perspective for these objectives and suggests the connection with TMPD in the case of linear inverse problems, as discussed in Sec. 4.2.

## 4.2 EXACT SOLUTION FOR LINEAR INVERSE PROBLEMS

As discussed in Sec. 3.2, the *local MAP solution matches the posterior mean* for Gaussian posteriors $p(x_t \mid x_0, y)$ arising from linear inverse problems $p(y \mid x_0) = \mathcal{N}(Hx_0, \sigma_y^2 \mathbb{I})$ with a Gaussian assumption on the prior $p(x_0 \mid x_t) = \mathcal{N}(x_0; m_{0|t}, \Sigma_{0|t})$. Solving in closed form for the posterior mean as in (Boys et al., 2023), we have

$$x_0^* = m_{0|t} + \Sigma_{0|t} H^T (H \Sigma_{0|t} H^T + \sigma_y^2 \mathbb{I})^{-1}(y - H m_{0|t}).$$
$$= m_{0|t} + \frac{\sigma_t^2}{\alpha_t} \nabla_{x_t} \log p(y \mid x_t) \tag{17}$$

We recover Tweedie Moment-Projected Diffusion (Boys et al., 2023) as a special case for $\Sigma_{0|t}^{\text{TMPD}} = \frac{\sigma_t^2}{\alpha_t} \nabla_{x_t} m_{0|t}(x_t)$, which is expensive since it requires the gradient with respect to the denoiser $m_{0|t}$. Thus, Local MAP Sampling reduced to DPS.

When applying LMAPS to linear inverse problems, we assume $\Sigma_{0|t} = \frac{k}{\text{SNR}_t}\mathbb{I}$ as in Sec. 4.1, and optimize with $K$ steps of gradient descent at each timestep despite the availability of the closed form in Eq. (17). We include solving LMAPS with analytical solution in App. E.3.

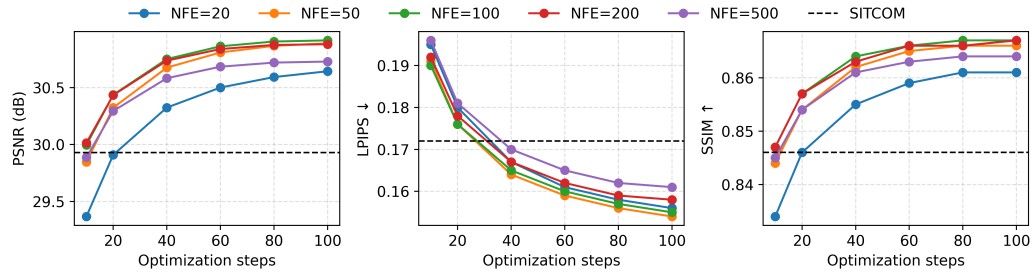

Figure 4: Ablation study on optimization steps vs. diffusion steps (NFEs) for Gaussian Deblurring.

## 5 EXPERIMENTS

### 5.1 EXPERIMENTAL SETUP

**Inverse problems**. We evaluate our method on image restoration and scientific inverse problems. For linear image restoration, we consider (1) super-resolution, (2) Gaussian deblurring, (3) motion deblurring, (4) inpainting (with a box mask), and (5) inpainting (with a 70% random mask). For nonlinear image restoration, we consider (1) phase retrieval, (2) high dynamic range (HDR) reconstruction, (3) nonlinear deblurring, (4) JPEG restoration, (5) quantization, where HDR, JPEG restoration and quantization are nonlinear inverse problems with non-differentiable operators. For scientific inverse problems, we adopt the benchmark from InverseBench (Zheng et al., 2025), which includes Linear Inverse Scattering (LIS), Compressed sensing MRI (CS-MRI) and Black Hole Imaging. More details are provided in the App. D.

**Dataset and Pretrained models**. For image restoration, we evaluated our method on FFHQ (Karras et al., 2019) $256 \times 256$ and ImageNet $256 \times 256$ datasets (Deng et al., 2009). Following DAPS, we test the same subset of 100 images for both datasets. For scientific inverse problems, we adopt the same dataset as InverseBench (Zheng et al., 2025). For image restoration tasks, we utilize the pre-trained checkpoint (Chung et al., 2022) on the FFHQ dataset and the pre-trained checkpoint (Dhariwal & Nichol, 2021) on the ImageNet dataset. For scientific inverse problems, we adopt the pre-trained checkpoints from InverseBench.

**Baselines**. We compare our method with the following baselines: DDNM (Wang et al., 2022), DDRM (Kawar et al., 2022), ΠGDM (Song et al., 2023b), DPS (Chung et al., 2022), LGD (Song et al., 2023c), PnP-DM (Wu et al., 2024), FPS (Dou & Song, 2024), MCG-diff (Cardoso et al., 2023), RedDiff (Mardani et al., 2023), DAPS (Zhang et al., 2025a), DiffPIR (Zhu et al., 2023), DCDP (Li et al., 2024), SITCOM (Alkhouri et al., 2024), DMPlug (Wang et al., 2024), MGDM (Janati et al., 2025), MAP-GA (Gutha et al., 2025), MMPS (Rozet et al., 2024).

**Metrics**. For image restoration tasks, we report Peak Signal-to-Noise Ratio (PSNR), Structural SIMilarity Index (SSIM), and Learned Perceptual Image Patch Similarity (LPIPS) (Zhang et al., 2018). For scientific inverse problems, we primarily present PSNR in the main text, while additional task-specific metrics are provided in the App. E.

### 5.2 MAIN RESULTS

**Ablation studies**. Figure 4 presents ablation studies on optimization steps across different diffusion steps. The best performance is typically observed at NFE = 200–500, where increasing the number of optimization steps per diffusion step. yields notable improvements. Compared to the baseline SITCOM (600 NFEs with gradient computation through the U-Net), LMAPS attains similar performance while requiring substantially fewer computational resources. We report runtime comparisons for various methods on Deblurring task in Table 3 (App. C).

**Image restoration**. In Table 1, we present quantitative results for image restoration tasks on FFHQ and ImageNet datasets. The table covers 10 tasks, 3 restoration quality metrics, and 2 datasets, totaling 60 results. LMAPS achieves the best performance in 43 out of 60 cases. Generally, LMAPS demonstrates superior performance than DAPS for most of the tasks with less computational cost. LMAPS improves $> 2$dB PSNR across motion deblurring, JPEG restoration and quantization tasks.

**Scientific inverse problems**. In Table 2, we report quantitative results of solving scientific inverse problems: Linear Inverse Scattering (LIS), CS-MRI, Black Hole Imaging. LMAPS demonstrates the best PSNR across all tasks, improved more than 1.5 dB PSNR for 3 LIS tasks.

Table 1: Quantitative evaluation of solving image restoration FFHQ (left) and ImageNet (right), with Gaussian noise ($\sigma_y = 0.05$): 5 linear and 5 nonlinear tasks (3 non-differentiable). Results are reported as mean PSNR, SSIM, and LPIPS across 100 images. Best results are highlighted in bold. For phase retrieval, DAPS and LMAPS select the best result from 4 runs for each image.

| Task | Method | FFHQ PSNR ↑ | SSIM ↑ | LPIPS ↓ | ImageNet PSNR ↑ | SSIM ↑ | LPIPS ↓ |
|---|---|---|---|---|---|---|---|
| SR 4× | DPS | 25.86 | 0.753 | 0.269 | 21.13 | 0.489 | 0.361 |
| | DDRM | 26.58 | 0.782 | 0.282 | 22.62 | 0.521 | 0.324 |
| | DDNM | 28.03 | 0.795 | 0.197 | 23.96 | 0.604 | 0.475 |
| | DCDP | 28.66 | 0.807 | 0.178 | – | – | – |
| | FPS-SMC | 28.42 | 0.813 | 0.204 | 24.82 | 0.703 | 0.313 |
| | DiffPIR | 26.64 | – | 0.260 | 23.18 | – | 0.371 |
| | DAPS | 29.07 | 0.818 | 0.177 | 25.89 | 0.694 | 0.276 |
| | DMPlug | 28.86 | 0.820 | 0.128 | – | – | – |
| | MMPS | 28.45 | 0.811 | 0.106 | – | – | – |
| | SITCOM | 30.55 | 0.864 | 0.154 | **27.07** | **0.746** | **0.228** |
| | MGDM | 27.81 | 0.798 | 0.111 | 25.44 | 0.684 | 0.246 |
| | MAP-GA | 29.97 | 0.844 | 0.178 | 26.00 | 0.708 | 0.267 |
| | LMAPS | **30.74** | **0.869** | 0.165 | 26.72 | 0.739 | 0.242 |
| Inpaint (Box) | DPS | 22.51 | 0.792 | 0.209 | 18.94 | 0.722 | 0.257 |
| | DDRM | 22.26 | 0.801 | 0.207 | 18.63 | 0.733 | 0.254 |
| | DDNM | 24.47 | 0.837 | 0.235 | 21.64 | 0.748 | 0.319 |
| | DCDP | 23.89 | 0.760 | 0.163 | – | – | – |
| | FPS-SMC | 24.86 | 0.823 | 0.146 | **22.16** | 0.726 | 0.208 |
| | DAPS | 24.07 | 0.814 | 0.133 | 21.43 | 0.725 | 0.214 |
| | SITCOM | 24.95 | 0.849 | 0.131 | 19.72 | 0.784 | **0.164** |
| | MMPS | 23.38 | 0.853 | **0.084** | – | – | – |
| | MAP-GA | 24.77 | 0.850 | 0.123 | 20.71 | 0.802 | 0.198 |
| | LMAPS | **25.17** | **0.876** | 0.108 | 21.25 | 0.803 | 0.204 |
| Inpaint (Random) | DPS | 25.46 | 0.823 | 0.203 | 23.52 | 0.745 | 0.297 |
| | DDNM | 29.91 | 0.817 | 0.121 | **31.16** | 0.841 | 0.191 |
| | DCDP | 30.69 | 0.842 | 0.142 | – | – | – |
| | FPS-SMC | 28.21 | 0.823 | 0.261 | 24.52 | 0.701 | 0.316 |
| | DAPS | 31.12 | 0.844 | 0.098 | 28.44 | 0.775 | 0.135 |
| | SITCOM | 33.96 | 0.928 | 0.082 | 29.74 | 0.855 | 0.115 |
| | DMPlug | 31.55 | 0.892 | 0.110 | – | – | – |
| | MMPS | 31.91 | 0.905 | **0.041** | – | – | – |
| | MAP-GA | 32.00 | 0.908 | 0.088 | 28.09 | 0.830 | 0.143 |
| | LMAPS | **34.51** | **0.938** | 0.066 | 30.59 | **0.876** | **0.100** |
| Gaussian Deblurring | DPS | 25.87 | 0.764 | 0.219 | 20.31 | 0.598 | 0.397 |
| | DDRM | 24.93 | 0.732 | 0.239 | 21.26 | 0.564 | 0.443 |
| | DCDP | 27.50 | 0.699 | 0.304 | – | – | – |
| | FPS-SMC | 26.54 | 0.773 | 0.253 | 23.91 | 0.601 | 0.387 |
| | DiffPIR | 27.36 | – | 0.236 | 22.80 | – | 0.355 |
| | DAPS | 29.19 | 0.817 | 0.165 | 26.15 | 0.684 | 0.253 |
| | SITCOM | 29.93 | 0.846 | 0.172 | 26.39 | 0.716 | 0.260 |
| | MGDM | 27.78 | 0.791 | **0.110** | 25.50 | 0.682 | 0.289 |
| | LMAPS | **30.88** | **0.867** | 0.158 | **26.65** | **0.727** | **0.250** |
| Motion Deblurring | DPS | 24.52 | 0.801 | 0.246 | 18.96 | 0.629 | 0.423 |
| | DCDP | 25.08 | 0.512 | 0.364 | – | – | – |
| | FPS-SMC | 27.39 | 0.826 | 0.227 | 24.52 | 0.647 | 0.326 |
| | DiffPIR | 26.57 | – | 0.255 | 24.01 | – | 0.366 |
| | DAPS | 29.66 | 0.847 | 0.157 | 27.86 | 0.766 | 0.196 |
| | SITCOM | 29.36 | 0.840 | 0.185 | 26.76 | 0.746 | 0.242 |
| | MMPS | 31.15 | 0.870 | **0.075** | – | – | – |
| | MGDM | 26.72 | 0.776 | 0.124 | 24.52 | 0.659 | 0.278 |
| | LMAPS | **32.62** | **0.902** | 0.117 | **28.42** | **0.796** | **0.204** |
| Phase Retrieval | DPS | $17.64_{\pm2.97}$ | $0.441_{\pm0.129}$ | $0.410_{\pm0.090}$ | $16.81_{\pm3.61}$ | $0.427_{\pm0.143}$ | $0.447_{\pm0.099}$ |
| | RED-diff | $15.60_{\pm4.48}$ | $0.398_{\pm0.195}$ | $0.596_{\pm0.092}$ | $14.98_{\pm3.75}$ | $0.386_{\pm0.057}$ | $0.536_{\pm0.129}$ |
| | MGDM | $19.24_{\pm8.22}$ | $0.533_{\pm0.271}$ | $0.346_{\pm0.254}$ | $13.77_{\pm4.30}$ | $0.293_{\pm0.196}$ | $0.578_{\pm0.169}$ |
| | DAPS | $30.63_{\pm3.13}$ | $0.851_{\pm0.072}$ | $0.139_{\pm0.060}$ | $21.39_{\pm6.59}$ | $0.473_{\pm0.226}$ | $0.372_{\pm0.166}$ |
| | LMAPS | $\mathbf{31.56_{\pm3.02}}$ | $\mathbf{0.867_{\pm0.057}}$ | $\mathbf{0.126_{\pm0.052}}$ | $\mathbf{22.86_{\pm7.50}}$ | $\mathbf{0.596_{\pm0.267}}$ | $\mathbf{0.313_{\pm0.176}}$ |
| Nonlinear Deblurring | DPS | $23.39_{\pm2.01}$ | $0.263_{\pm0.082}$ | $0.278_{\pm0.060}$ | $22.49_{\pm3.20}$ | $0.591_{\pm0.101}$ | $0.306_{\pm0.081}$ |
| | RED-diff | $30.86_{\pm0.51}$ | $0.795_{\pm0.028}$ | $0.160_{\pm0.034}$ | $\mathbf{30.07_{\pm1.41}}$ | $0.754_{\pm0.023}$ | $0.211_{\pm0.083}$ |
| | DCDP | $27.92_{\pm2.64}$ | $0.779_{\pm0.067}$ | $0.183_{\pm0.051}$ | – | – | – |
| | DAPS | $28.29_{\pm1.77}$ | $0.783_{\pm0.036}$ | $0.155_{\pm0.032}$ | $27.73_{\pm3.23}$ | $0.724_{\pm0.048}$ | $0.169_{\pm0.056}$ |
| | DMPlug | $27.65_{\pm2.98}$ | $0.795_{\pm0.080}$ | $0.181_{\pm0.056}$ | – | – | – |
| | SITCOM | $29.19_{\pm2.35}$ | $0.785_{\pm0.093}$ | $0.190_{\pm0.014}$ | $28.55_{\pm3.87}$ | $\mathbf{0.798_{\pm0.092}}$ | $\mathbf{0.149_{\pm0.050}}$ |
| | MGDM | $23.88_{\pm2.61}$ | $0.664_{\pm0.081}$ | $0.271_{\pm0.085}$ | $22.63_{\pm2.98}$ | $0.583_{\pm0.122}$ | $0.394_{\pm0.117}$ |
| | LMAPS | $29.93_{\pm1.83}$ | $\mathbf{0.855_{\pm0.035}}$ | $\mathbf{0.150_{\pm0.034}}$ | $28.03_{\pm3.62}$ | $0.774_{\pm0.099}$ | $0.183_{\pm0.065}$ |
| High Dynamic Range | DPS | $22.73_{\pm6.07}$ | $0.591_{\pm0.141}$ | $0.264_{\pm0.156}$ | $19.23_{\pm2.52}$ | $0.582_{\pm0.082}$ | $0.503_{\pm0.106}$ |
| | DAPS | $27.12_{\pm3.53}$ | $0.752_{\pm0.041}$ | $0.162_{\pm0.072}$ | $26.30_{\pm4.10}$ | $0.717_{\pm0.067}$ | $0.175_{\pm0.107}$ |
| | SITCOM | $28.02_{\pm3.28}$ | $0.812_{\pm0.108}$ | $0.174_{\pm0.081}$ | $25.59_{\pm3.66}$ | $0.170_{\pm0.141}$ | $0.198_{\pm0.177}$ |
| | MGDM | $25.73_{\pm4.28}$ | $0.796_{\pm0.151}$ | $\mathbf{0.100_{\pm0.096}}$ | $23.43_{\pm4.68}$ | $0.754_{\pm0.165}$ | $0.173_{\pm0.152}$ |
| | LMAPS | $\mathbf{28.87_{\pm3.39}}$ | $\mathbf{0.884_{\pm0.082}}$ | $0.141_{\pm0.074}$ | $\mathbf{27.02_{\pm4.00}}$ | $\mathbf{0.860_{\pm0.096}}$ | $0.158_{\pm0.090}$ |
| JPEG Restoration (QF=5) | ΠGDM | $25.04_{\pm1.28}$ | $0.755_{\pm0.060}$ | $0.270_{\pm0.045}$ | $22.41_{\pm2.23}$ | $0.606_{\pm0.144}$ | $0.417_{\pm0.087}$ |
| | LMAPS | $\mathbf{27.25_{\pm1.37}}$ | $\mathbf{0.814_{\pm0.045}}$ | $\mathbf{0.260_{\pm0.043}}$ | $\mathbf{24.96_{\pm2.46}}$ | $\mathbf{0.703_{\pm0.124}}$ | $\mathbf{0.340_{\pm0.089}}$ |
| Quantization | ΠGDM | $25.82_{\pm1.29}$ | $0.789_{\pm0.063}$ | $0.255_{\pm0.046}$ | $22.34_{\pm2.26}$ | $0.425_{\pm0.110}$ | $0.605_{\pm0.156}$ |
| | LMAPS | $\mathbf{29.51_{\pm1.14}}$ | $\mathbf{0.844_{\pm0.467}}$ | $\mathbf{0.229_{\pm0.474}}$ | $\mathbf{26.92_{\pm2.25}}$ | $\mathbf{0.748_{\pm0.114}}$ | $\mathbf{0.307_{\pm0.099}}$ |

Table 2: Quantitative evaluation of solving scientific inverse problems is conducted using PSNR as the evaluation metric. The tasks include: (i) three LIS settings with different numbers of receivers (NR = 360, 180, 60); (ii) four CS-MRI settings with varying subsampling ratios ($4\times$, $8\times$) and measurement types (noiseless and raw); and (iii) three Black Hole Imaging settings with different observation time ratios (3%, 10%, 100%).

| Method | LIS | | | CS-MRI | | | | Black Hole | | |
|---|---|---|---|---|---|---|---|---|---|---|
| | NR=360 | NR=180 | NR=60 | $4\times$ noiseless | $4\times$ raw | $8\times$ noiseless | $8\times$ raw | 100% | 10% | 3% |
| DDRM | 32.13 | 28.08 | 20.44 | – | – | – | – | – | – | – |
| DDNM | 26.28 | 35.02 | 29.24 | – | – | – | – | – | – | – |
| ΠGDM | 27.93 | 26.40 | 20.07 | – | – | – | – | – | – | – |
| DPS | 32.06 | 31.80 | 27.37 | 26.13 | 25.83 | 20.82 | 23.00 | 25.86 | 24.36 | 24.20 |
| LGD | 27.90 | 27.84 | 20.49 | – | – | – | – | 21.22 | 22.08 | 22.51 |
| DiffPIR | 34.24 | 34.01 | 26.32 | 28.31 | 27.60 | 26.78 | 26.26 | 25.01 | 23.84 | 24.12 |
| PnP-DM | 33.94 | 31.82 | 24.72 | 31.80 | 27.62 | 29.33 | 25.28 | 26.07 | 24.57 | 24.25 |
| DAPS | 34.64 | 33.16 | 25.88 | 31.48 | 28.61 | 29.01 | 27.10 | 25.60 | 23.99 | 23.54 |
| RED-diff | 36.56 | 35.41 | 27.07 | 29.36 | 28.71 | 26.76 | 27.33 | 23.77 | 22.53 | 20.74 |
| FPS | 33.24 | 29.62 | 21.32 | – | – | – | – | – | – | – |
| MCG-diff | 30.94 | 28.06 | 21.00 | – | – | – | – | – | – | – |
| LMAPS | **38.07** | **37.19** | **30.75** | **32.83** | **28.77** | **30.50** | **27.43** | **26.79** | **24.83** | **24.66** |

# 6 RELATED WORK

Recent advances in conditional generation have led to breakthroughs in text-to-image synthesis, semantic editing, and domain-specific applications such as image-to-image translation and controlled signal reconstruction (Song et al., 2023c; Ye et al., 2024; Skreta et al., 2025; Singhal et al., 2025; Zheng et al., 2023). These methods have been especially impactful in solving inverse problems, including image restoration and scientific reconstruction tasks (Wang et al., 2022; Zheng et al., 2025). A wide range of approaches have been developed, spanning linear projection methods (Wang et al., 2022; Kawar et al., 2022; Zhang et al., 2025b; Dou & Song, 2024), Monte Carlo sampling (Wu et al., 2023; Phillips et al., 2024), variational inference (Feng et al., 2023; Mardani et al., 2023; Janati et al., 2024), and optimization-based strategies (Song et al., 2023a; Zhu et al., 2023; Li et al., 2024; Wang et al., 2024; Alkhouri et al., 2024; He et al., 2023).

Among these, Diffusion Posterior Sampling (DPS) and its variants (Zhang et al., 2025a; Chung et al., 2022; Song et al., 2023c; Yu et al., 2023; Rout et al., 2024; Yang et al., 2024; Bansal et al., 2023; Boys et al., 2023; Song et al., 2023b; Ho & Salimans, 2022) have gained wide adoption due to their strong empirical performance and interpretability, as they directly sample from the posterior distribution $p(x_0 \mid y)$. More recently, attention has shifted toward maximum a posteriori (MAP) estimation with diffusion priors. Xu et al. (2025) argued that DPS is in fact more consistent with the principle of MAP estimation rather than true posterior sampling, although their proposed sampling algorithm differs from ours. Finally, Gutha et al. (2025) proposed sampling from the global MAP solution, $\arg\max p(x_0 \mid y)$, though their approach is largely restricted to linear inverse problems.

# 7 CONCLUSION

We presented Local MAP Sampling (LMAPS), a new inference framework that that iteratively solves local maximum-a-posteriori subproblems along the diffusion trajectory. By introducing a principled covariance approximation, an objective reformulation, and a gradient strategy for non-differentiable operators, LMAPS provides both theoretical clarity and practical effectiveness. Experiments across diverse image restoration and scientific inverse problems show that LMAPS consistently improves reconstruction quality, particularly on challenging tasks such as Box Inpainting, Phase Retrieval, JPEG restoration, and HDR.

**Future work.** In Bayesian inference, the global MAP plays a critical role and offers valuable insights contrasted with posterior sampling. Yet its utility has been largely overlooked, and efficiently solving the global MAP with diffusion priors remains an open challenge. Advancing in this direction could enable more probable reconstructions and make contributions to solving inverse problems.

## REPRODUCIBILITY STATEMENT

All code and instructions necessary to reproduce our experiments are anonymously available at `https://anonymous.4open.science/r/maps_inverse-BFCF`.

## ETHICS STATEMENT

This work does not present any foreseeable ethical issues.

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

# A    GAUSSIAN MIXTURE TOY EXAMPLE

To gain intuition about posterior mean and MAP estimates in diffusion models, we consider a tractable toy prior $\pi_0(x_0)$ given by a Gaussian mixture:

$$\pi_0(x_0) = \sum_{k=1}^{K} \pi_k \, \mathcal{N}(x_0; \, \mu_k, \Sigma_k), \tag{18}$$

where $\pi_k > 0$ and $\sum_k \pi_k = 1$.

**Forward kernel.**    As in the unconditional diffusion model, the forward corruption is

$$p_t(x_t \mid x_0) = \mathcal{N}(x_t; \, \alpha_t x_0, \sigma_t^2 I). \tag{19}$$

Thus the marginal $p_t(x_t) = \int p_t(x_t \mid x_0) \, \pi_0(x_0) \, dx_0$ is itself a Gaussian mixture.

**Posterior distribution.**    By Bayes' rule,

$$p(x_0 \mid x_t) \propto p_t(x_t \mid x_0) \, \pi_0(x_0). \tag{20}$$

Conditioned on mixture component $k$, the posterior remains Gaussian:

$$p(x_0 \mid x_t, k) = \mathcal{N}(x_0; \, m_k, S_k), \tag{21}$$

$$S_k = \left( \Sigma_k^{-1} + \frac{\alpha_t^2}{\sigma_t^2} I \right)^{-1}, \tag{22}$$

$$m_k = S_k \left( \Sigma_k^{-1} \mu_k + \frac{\alpha_t}{\sigma_t^2} x_t \right). \tag{23}$$

The responsibilities are

$$r_k(x_t) = \frac{\pi_k \, \mathcal{N}(x_t; \, \alpha_t \mu_k, \, \alpha_t^2 \Sigma_k + \sigma_t^2 I)}{\sum_j \pi_j \, \mathcal{N}(x_t; \, \alpha_t \mu_j, \, \alpha_t^2 \Sigma_j + \sigma_t^2 I)}. \tag{24}$$

Hence the full posterior is itself a Gaussian mixture:

$$p(x_0 \mid x_t) = \sum_{k=1}^{K} r_k(x_t) \, \mathcal{N}(x_0; \, m_k, S_k). \tag{25}$$

**Posterior mean.**    The ideal denoiser in this case has a closed form:

$$m_{0|t}(x_t) := \mathbb{E}[x_0 \mid x_t] = \sum_{k=1}^{K} r_k(x_t) \, m_k. \tag{26}$$

For a fixed $x_t$, the posterior mean is a responsibility-weighted average of the component-wise posterior means, and can fall between mixture modes when the conditional posterior is multimodal.

**Local MAP.**    Each component posterior has its mode at $m_k$. A local MAP predictor is obtained by selecting the component with the highest posterior peak density,

$$k^\star(x_t) = \arg\max_k \frac{r_k(x_t)}{\sqrt{(2\pi)^d \det S_k}}, \qquad x_0^*(t, x_t) = m_{k^\star(x_t)}. \tag{27}$$

Unlike the posterior mean, this estimate is *mode-seeking* and stays in high-density regions.

**DDIM iteration.**    Replacing the generic denoiser $m_{0|t}(x_t)$ in the DDIM update with either the posterior mean, local MAP yields 2 distinct variants of the reverse process:

$$x_{t-\Delta t} = g(m_{0|t}(x_t), \, x_t, \epsilon) \qquad \text{(posterior mean)} \tag{28}$$

$$x_{t-\Delta t} = g(x_0^*(t, x_t), \, x_t, \epsilon) \qquad \text{(local MAP)} \tag{29}$$

This toy setup makes explicit the distinction between *mean-based* denoising and *MAP-based* denoising.

**Bias toward high-posterior modes.** The above construction allows us to make precise in which sense local MAP is biased toward high-density regions of the posterior. For clarity, consider the special case where all mixture components share the same covariance, $\Sigma_k = \Sigma$, so that $S_k$ and $\det S_k$ are independent of $k$. In this setting, the local MAP predictor simplifies to

$$k^\star(x_t) = \arg\max_k \ r_k(x_t), \qquad x_0^*(t, x_t) = m_{k^\star(x_t)}, \tag{30}$$

that is, it selects the component with the largest responsibility. Equivalently, $x_0^*(t, x_t)$ is the maximizer of the joint posterior over the discrete–continuous pair $(k, x_0)$,

$$(k^\star, x_0^\star) = \arg\max_{k, x_0} \ p(x_0, k \mid x_t) = \arg\max_k \ p(k \mid x_t), \tag{31}$$

where the maximizer over $x_0$ within each component is $m_k$. Thus local MAP coincides with the MAP estimator of the latent mixture index $k$ (under 0–1 loss), followed by the corresponding component-wise posterior mode $m_k$.

Let $\mathcal{R}_k = \{x_t : k^\star(x_t) = k\}$ denote the region of the diffusion state space where component $k$ is selected. If we draw $x_t \sim p_t(x_t)$ and then apply local MAP, the probability that LMAPS outputs a sample associated with component $k$ is

$$q_t(k) := \mathbb{P}\big[k^\star(x_t) = k\big] = \int_{\mathcal{R}_k} p_t(x_t)\,dx_t. \tag{32}$$

By definition of $\mathcal{R}_k$, each $x_t \in \mathcal{R}_k$ satisfies $r_k(x_t) \geq r_j(x_t)$ for all $j \neq k$, so $\mathcal{R}_k$ collects those diffusion states where component $k$ dominates the posterior responsibilities. Consequently, $q_t(k)$ is concentrated on modes with large posterior weight: whenever a component has small responsibilities $r_k(x_t)$ for almost all $x_t$, its region $\mathcal{R}_k$ has small measure and $q_t(k)$ is correspondingly small.

In the well-separated mixture regime, where the means $\{\mu_k\}$ are far apart relative to $\Sigma$ and the diffusion noise, the posterior responsibilities $r_k(x_t)$ are nearly 0–1 valued. In this case, each region $\mathcal{R}_k$ is essentially the basin of attraction of mode $k$, and

$$q_t(k) \approx \int_{\text{basin}(k)} p_t(x_t)\,dx_t, \tag{33}$$

which is dominated by components with the highest posterior mass. Thus, even though LMAPS does not sample from the exact posterior mixture $\sum_k r_k(x_t)\mathcal{N}(m_k, S_k)$, its outputs are systematically biased toward high-posterior modes and avoid low-density regions between them. In contrast, posterior mean denoising yields mode-averaging estimates that may lie in low-density areas, and local posterior sampling explores all mixture components proportionally to their posterior mass. This toy example therefore formalizes the intuition that LMAPS interpolates between global MAP and posterior sampling by producing samples that concentrate on highly likely regions of the posterior while remaining stochastic along the diffusion trajectory.

## B   POSTERIOR COVARIANCE AND ASYMPTOTIC ISOTROPY

**Proposition 1** (Gaussian prior)**.** *Consider the forward noising process*

$$x_t = \alpha_t x_0 + \sigma_t \epsilon, \qquad \epsilon \sim \mathcal{N}(0, \mathbb{I}), \tag{34}$$

*and a Gaussian prior $x_0 \sim \mathcal{N}(\mu_0, \Sigma_0)$ with $\Sigma_0 \succ 0$. Then the posterior $p(x_0 \mid x_t)$ is Gaussian with covariance*

$$\Sigma_{0|t} = \big(\Sigma_0^{-1} + \tfrac{\alpha_t^2}{\sigma_t^2}\mathbb{I}\big)^{-1} \preceq \frac{\sigma_t^2}{\alpha_t^2}\,\mathbb{I}. \tag{35}$$

*Moreover, as $\sigma_t^2/\alpha_t^2 \to 0$, the covariance admits the asymptotic expansion*

$$\Sigma_{0|t} = \frac{\sigma_t^2}{\alpha_t^2}\,\mathbb{I} + \mathcal{O}\big((\tfrac{\sigma_t^2}{\alpha_t^2})^2\big), \tag{36}$$

*so the leading term is isotropic and any anisotropy appears only in higher–order corrections.*

*Proof.* Since $(x_0, x_t)$ is jointly Gaussian under the model

$$x_0 \sim \mathcal{N}(\mu_0, \Sigma_0), \qquad x_t \mid x_0 \sim \mathcal{N}(\alpha_t x_0, \sigma_t^2 \mathbb{I}), \tag{37}$$

the posterior $p(x_0 \mid x_t)$ is Gaussian. Equivalently, we can view $x_t$ as a linear observation of $x_0$ with observation matrix $H = \alpha_t \mathbb{I}$ and noise covariance $R = \sigma_t^2 \mathbb{I}$. The standard linear Gaussian posterior formula (or Kalman update) yields

$$\Sigma_{0|t}^{-1} = \Sigma_0^{-1} + H^\top R^{-1} H = \Sigma_0^{-1} + \frac{\alpha_t^2}{\sigma_t^2} \mathbb{I}, \tag{38}$$

so

$$\Sigma_{0|t} = \left(\Sigma_0^{-1} + \frac{\alpha_t^2}{\sigma_t^2} \mathbb{I}\right)^{-1}. \tag{39}$$

For the Loewner-order upper bound, note that $\Sigma_0^{-1} \succeq 0$, so

$$\Sigma_0^{-1} + \frac{\alpha_t^2}{\sigma_t^2} \mathbb{I} \succeq \frac{\alpha_t^2}{\sigma_t^2} \mathbb{I}. \tag{40}$$

For positive definite matrices, the matrix inverse is order-reversing: if $A \succeq B \succ 0$, then $A^{-1} \preceq B^{-1}$. Applying this with $A = \Sigma_0^{-1} + \frac{\alpha_t^2}{\sigma_t^2} \mathbb{I}$ and $B = \frac{\alpha_t^2}{\sigma_t^2} \mathbb{I}$ gives

$$\Sigma_{0|t} = A^{-1} \preceq B^{-1} = \frac{\sigma_t^2}{\alpha_t^2} \mathbb{I}. \tag{41}$$

For the asymptotic expansion, factor out the isotropic term:

$$\Sigma_{0|t} = \left(\Sigma_0^{-1} + \frac{\alpha_t^2}{\sigma_t^2} \mathbb{I}\right)^{-1} = \frac{\sigma_t^2}{\alpha_t^2} \left(\mathbb{I} + \frac{\sigma_t^2}{\alpha_t^2} \Sigma_0^{-1}\right)^{-1}. \tag{42}$$

Let $\varepsilon_t := \frac{\sigma_t^2}{\alpha_t^2}$. For $\varepsilon_t \to 0$ we may use the Neumann series

$$\left(\mathbb{I} + \varepsilon_t \Sigma_0^{-1}\right)^{-1} = \mathbb{I} - \varepsilon_t \Sigma_0^{-1} + \mathcal{O}(\varepsilon_t^2), \tag{43}$$

which yields

$$\Sigma_{0|t} = \varepsilon_t \left(\mathbb{I} - \varepsilon_t \Sigma_0^{-1} + \mathcal{O}(\varepsilon_t^2)\right) = \varepsilon_t \mathbb{I} + \mathcal{O}(\varepsilon_t^2) = \frac{\sigma_t^2}{\alpha_t^2} \mathbb{I} + \mathcal{O}\left(\left(\frac{\sigma_t^2}{\alpha_t^2}\right)^2\right). \tag{44}$$

The leading term is therefore isotropic, and any anisotropy is of order $\left(\frac{\sigma_t^2}{\alpha_t^2}\right)^2$. $\qquad \square$

**Proposition 2** (General prior and asymptotic isotropy). *Assume the forward noising process*

$$x_t = \alpha_t x_0 + \sigma_t \epsilon, \qquad \epsilon \sim \mathcal{N}(0, \mathbb{I}), \tag{45}$$

*and an arbitrary prior density $p(x_0)$ such that $-\log p(x_0)$ is twice continuously differentiable. Then the negative log-posterior is*

$$-\log p(x_0 \mid x_t) = -\log p(x_0) + \frac{1}{2\sigma_t^2} \|x_t - \alpha_t x_0\|^2 + const, \tag{46}$$

*and its Hessian with respect to $x_0$ satisfies*

$$\nabla_{x_0}^2 \left[-\log p(x_0 \mid x_t)\right] = \frac{\alpha_t^2}{\sigma_t^2} \mathbb{I} + H_{\text{prior}}(x_0), \tag{47}$$

*where $H_{\text{prior}}(x_0) := \nabla_{x_0}^2 \left[-\log p(x_0)\right]$. If $H_{\text{prior}}(x_0)$ is bounded in operator norm on the region of interest, then as $\sigma_t^2 \to 0$ (and $\alpha_t \to 1$), the local Gaussian (Laplace) approximation to $p(x_0 \mid x_t)$ has covariance*

$$\Sigma_{0|t}^{\text{Laplace}}(x_0) = \frac{\sigma_t^2}{\alpha_t^2} \mathbb{I} + \mathcal{O}\left(\left(\frac{\sigma_t^2}{\alpha_t^2}\right)^2\right), \tag{48}$$

*and is therefore asymptotically isotropic as $t \to 0$.*

Table 3: Sampling time of LMAPS on Deblurring tasks with FFHQ 256. The non-parallel single-image sampling time on the FFHQ 256 dataset using one NVIDIA A6000 GPU. NFE refers to diffusion timesteps, while optimization steps refer to inner loop optimizations in respective methods.

| Configuration | ODE Steps | Optimization Steps | NFE | Second/Image | LPIPS |
|---|---|---|---|---|---|
| DAPS | 5 | 100 | 200 | 110 | 0.165 |
| SITCOM | – | 30 | 600 | 73 | 0.172 |
| DPS | – | – | 1000 | 138 | 0.219 |
| | – | 100 | 200 | 61 | 0.158 |
| | – | 10 | 100 | 6 | 0.190 |
| MAPS | – | 100 | 20 | 6 | 0.156 |
| | – | 20 | 100 | 6 | 0.176 |
| | – | 20 | 20 | 2 | 0.180 |

*Proof.* The expression for $-\log p(x_0 \mid x_t)$ follows directly from Bayes' rule and the Gaussian likelihood:

$$p(x_t \mid x_0) \propto \exp\left(-\frac{1}{2\sigma_t^2} \|x_t - \alpha_t x_0\|^2\right). \tag{49}$$

Taking the Hessian with respect to $x_0$ gives

$$\nabla_{x_0}^2 \left[\frac{1}{2\sigma_t^2} \|x_t - \alpha_t x_0\|^2\right] = \frac{\alpha_t^2}{\sigma_t^2} \mathbb{I}, \tag{50}$$

while the prior contributes

$$\nabla_{x_0}^2 \left[-\log p(x_0)\right] = H_{\text{prior}}(x_0). \tag{51}$$

Therefore,

$$H_{\text{post}}(x_0) := \nabla_{x_0}^2 \left[-\log p(x_0 \mid x_t)\right] = \frac{\alpha_t^2}{\sigma_t^2} \mathbb{I} + H_{\text{prior}}(x_0). \tag{52}$$

Assume $\|H_{\text{prior}}(x_0)\|_{\text{op}} \leq C$ for some constant $C$. Then, in the regime $\sigma_t^2 \to 0$ and $\alpha_t \to 1$, the dominant term in $H_{\text{post}}(x_0)$ is the isotropic matrix $\frac{\alpha_t^2}{\sigma_t^2}\mathbb{I}$. Define again $\varepsilon_t := \frac{\sigma_t^2}{\alpha_t^2}$ and write

$$H_{\text{post}}(x_0) = \frac{\alpha_t^2}{\sigma_t^2} \left(\mathbb{I} + \varepsilon_t H_{\text{prior}}(x_0)\right). \tag{53}$$

The local Gaussian (Laplace) approximation uses $\Sigma_{0|t}^{\text{Laplace}}(x_0) = H_{\text{post}}(x_0)^{-1}$. Applying the Neumann series to $\left(\mathbb{I} + \varepsilon_t H_{\text{prior}}(x_0)\right)^{-1}$ for small $\varepsilon_t$ yields

$$\left(\mathbb{I} + \varepsilon_t H_{\text{prior}}(x_0)\right)^{-1} = \mathbb{I} - \varepsilon_t H_{\text{prior}}(x_0) + \mathcal{O}(\varepsilon_t^2), \tag{54}$$

so

$$\Sigma_{0|t}^{\text{Laplace}}(x_0) = \varepsilon_t \left(\mathbb{I} - \varepsilon_t H_{\text{prior}}(x_0) + \mathcal{O}(\varepsilon_t^2)\right) = \varepsilon_t \mathbb{I} + \mathcal{O}(\varepsilon_t^2) = \frac{\sigma_t^2}{\alpha_t^2} \mathbb{I} + \mathcal{O}\left(\left(\frac{\sigma_t^2}{\alpha_t^2}\right)^2\right). \tag{55}$$

Thus the leading term of the local covariance is isotropic, and any anisotropy is of strictly higher order in $\sigma_t^2/\alpha_t^2$. $\qquad\square$

## C  SAMPLING EFFICIENCY

We present a comparison of sampling times among LMAPS, DAPS, and SITCOM. Among them, SITCOM and DAPS achieve the third- and second-best results, respectively, while LMAPS demonstrates the best performance with lower computation time.

# D EXPERIMENT DETAILS

## D.1 DATASET DETAILS

For scientific inverse problems, we adopt fluorescence microscopy images from InverseBench (Zheng et al., 2025) on linear inverse scattering tasks, General Relativistic MagnetoHydroDynamic (GRMHD) (Mizuno, 2022) on black hole imaging, and multi-coil raw $k$-space data from the fastMRI knee dataset (Zbontar et al., 2018) on CS-MRI.

## D.2 INVERSE PROBLEM DETAILS

**Baselines from DAPS** (Zhang et al., 2025a). For image restoration tasks include: (1) super-resolution, (2) Gaussian deblurring, (3) motion deblurring, (4) inpainting (with a box mask), and (5) inpainting (with a 70% random mask), (6) phase retrieval, (7) high dynamic range (HDR) reconstruction, (8) nonlinear deblurring, we follow the same experimental setup as in DAPS.

**InverseBench** (Zheng et al., 2025). For scientific inverse problems, we adopt the setting introduced in InverseBench.

**JPEG Restoration**. We address JPEG restoration with quality factors of $\mathrm{QF} = 5$.

**Quantization**. We model quantization by discretizing the measurement into $2^{n_{\text{bits}}}$ uniformly spaced levels. Formally, the forward operator is defined as

$$\mathcal{H}(x) = \frac{\lfloor x \cdot (2^{n_{\text{bits}}} - 1) + 0.5 \rfloor}{2^{n_{\text{bits}}} - 1},\tag{56}$$

which rounds the input $x$ to the nearest quantization level. In this work, we focus on the challenging case of 2-bit quantization, where only four distinct measurement levels are available, significantly reducing precision and making accurate reconstruction more difficult.

## D.3 BASELINE DETAILS

For SITCOM (Alkhouri et al., 2024), we use the hyperparameter configuration recommended in the original paper, with $N = 20$ and $K = 30$, resulting in $600$ NFEs and requiring gradient computation with respect to the U-Net.

For DMPlug (Wang et al., 2024), we set epoch $= 1000$ for SR, Inpainting (Random) and Nonlinear Deblurring, other parameters are the same as suggested in the original paper.

For MMPS (Rozet et al., 2024), we set steps as $100$, the maximum number of iterations $N = 5$.

For non-differentiable inverse problems, we use ΠGDM (Song et al., 2023b) as our baseline approaches, we adopt $NFE = 100$ as suggested in the original paper.

Other baselines we adopt the same reported results as in DAPS (Zhang et al., 2025a) and InverseBench (Zheng et al., 2025).

## D.4 COMPLETE LIST OF HYPER-PARAMETERS

We provide complete lost of hyper-paramers of LMAPS for different inverse problems in Table 4.

# E ADDITIONAL EXPERIMENT RESULTS

## E.1 SCIENTIFIC INVERSE PROBLEMS

We present additional evaluation metrics on linear inverse scattering in Table 5, compressed sensing MRI in Table 6, and black hole imaging in Table 7.

## E.2 ADDITIONAL VISUALIZATION

Additional visualization are presented in Figs. 5, 6, 7, 8, 9, 10, 11, 12, 13, 14.

| Tasks | Annealing step | Gradient step | Learning rate $\eta$ | $k_1$ | $k_2$ |
|---|---|---|---|---|---|
| SR $4\times$ | 200 | 100 | 0.05 | 0.15 | 20 |
| Inpaint (Box) | 200 | 100 | 0.02 | 0.5 | 50 |
| Inpaint (Random) | 200 | 100 | 0..01 | 0.22 | 100 |
| Gaussian Deblurring | 200 | 100 | 0.01 | 0.22 | 100 |
| Motion Deblurring | 200 | 100 | 0.01 | 0.25 | 100 |
| Phase Retrieval | 200 | 100 | 0.1 | 10 | 0.3 |
| Nonlinear Deblurring | 200 | 100 | 0.02 | 0.05 | 1 |
| High Dynamic Range | 200 | 100 | 0.04 | 0.2 | 10 |
| JPEG Restoration | 200 | 100 | 0.2 | 0.5 | 5 |
| Quantization | 200 | 20 | 0.2 | 0.5 | 5 |
| LIS (NR=360) | 200 | 50 | 1 | 0 | 5000 |
| LIS (NR=180) | 200 | 50 | 1 | 0 | 10000 |
| LIS (NR=60) | 200 | 50 | 1 | 0 | 30000 |
| CS-MRI ($4\times$, noiseless) | 200 | 100 | 0.01 | 0 | 100 |
| CS-MRI ($4\times$, raw) | 200 | 100 | 0.01 | 0.4 | 150 |
| CS-MRI ($8\times$, noiseless) | 200 | 100 | 0.01 | 0.4 | 150 |
| CS-MRI ($8\times$, raw) | 200 | 100 | 0.01 | 0.4 | 150 |
| Black Hole (ratio=100%) | 100 | 200 | 0.01 | 0.1 | 0.01 |
| Black Hole (ratio=10%) | 100 | 200 | 0.005 | 0.1 | 0.03 |
| Black Hole (ratio=3%) | 100 | 200 | 0.01 | 0.05 | 0.05 |

Table 4: Complete List of hyper-parameters of LMAPS for different inverse problems.

Table 5: Results on Linear inverse scattering. PSNR and SSIM of different algorithms on linear inverse scattering. Noise level $\sigma_y = 10^{-4}$.

| Number of receivers | 360 | | 180 | | 60 | |
|---|---|---|---|---|---|---|
| Methods | PSNR | SSIM | PSNR | SSIM | PSNR | SSIM |
| **Traditional** | | | | | | |
| FISTA-TV | 32.126 (2.139) | 0.979 (0.009) | 26.523 (2.678) | 0.914 (0.040) | 20.938 (2.513) | 0.709 (0.103) |
| **PnP diffusion prior** | | | | | | |
| DDRM | 32.598 (1.825) | 0.929 (0.012) | 28.080 (1.516) | 0.890 (0.019) | 20.436 (1.210) | 0.545 (0.037) |
| DDNM | 36.381 (1.098) | 0.935 (0.017) | 35.024 (0.993) | 0.895 (0.027) | 29.235 (3.376) | 0.917 (0.022) |
| IIGDM | 27.925 (3.211) | 0.889 (0.072) | 26.412 (3.430) | 0.816 (0.114) | 20.074 (2.608) | 0.540 (0.198) |
| DPS | 32.061 (2.163) | 0.846 (0.127) | 31.798 (2.163) | 0.862 (0.123) | 27.372 (3.415) | 0.813 (0.133) |
| LGD | 27.901 (2.346) | 0.812 (0.037) | 27.837 (3.031) | 0.803 (0.034) | 20.491 (3.031) | 0.552 (0.077) |
| DiffPIR | 34.241 (2.310) | 0.988 (0.006) | 34.010 (2.269) | 0.987 (0.006) | 26.321 (3.272) | 0.918 (0.028) |
| PnP-DM | 33.914 (2.054) | 0.988 (0.006) | 31.817 (2.073) | 0.981 (0.008) | 24.715 (2.874) | 0.909 (0.046) |
| DAPS | 34.641 (1.693) | 0.957 (0.006) | 33.160 (1.704) | 0.944 (0.009) | 25.875 (3.110) | 0.885 (0.030) |
| RED-diff | 36.556 (2.292) | 0.981 (0.005) | 35.411 (2.166) | 0.984 (0.004) | 27.072 (3.330) | 0.935 (0.037) |
| FPS | 33.242 (1.602) | 0.870 (0.026) | 29.624 (1.651) | 0.710 (0.040) | 21.323 (1.445) | 0.460 (0.030) |
| MCG-diff | 30.937 (1.964) | 0.751 (0.029) | 28.057 (1.672) | 0.631 (0.042) | 21.004 (1.571) | 0.445 (0.028) |
| LMAPS | **38.074** (1.905) | **0.994** (0.001) | **37.188** (1.815) | **0.990** (0.001) | **30.759** (3.539) | **0.967** (0.211) |

### E.3 COMPARISON BETWEEN ANALYTICAL SOLUTION AND GRADIENT DESCENT FOR SOLVING LMAPS

We present the comparison between analytical solution and gradient descent for solving LMAPS in Table 8. The results demonstrate that the analytical solution closely matches the gradient-descent-based optimizer, with only minor differences in reconstruction metrics. This confirms that our analytical formulation is a reliable and efficient approximation for solving the LMAPS objective.

### E.4 ADDITIONAL RESULTS ON NONLINEAR DEBLURRING

For Nonlinear Deblurring, the forward operator call is relatively expensive. The results on solving Nonlinear Deblurring with different annealing step and gradient step are shown in Table 9. LMPAPS can achieve competitive performance with only 100 annealing steps and 20 gradient steps.

Table 6: Results on compressed sensing MRI. Mean and standard deviation are reported over 94 test cases. Underline: the best across all methods. Bold: the best across PnP DM methods.

| Methods | ×4 Simulated (noiseless) | | | ×4 Raw | | | ×8 Simulated (noiseless) | | | ×8 Raw | | |
|---|---|---|---|---|---|---|---|---|---|---|---|---|
| | PSNR ↑ | SSIM ↑ | Data misfit ↓ | PSNR ↑ | SSIM ↑ | Data misfit ↓ | PSNR ↑ | SSIM ↑ | Data misfit ↓ | PSNR ↑ | SSIM ↑ | Data misfit ↓ |
| **Traditional** | | | | | | | | | | | | |
| Wavelet+$\ell_1$ | 29.45 (1.776) | 0.690 (0.121) | 0.306 (0.049) | 26.47 (1.508) | 0.598 (0.122) | 31.601 (15.286) | 25.97 (1.761) | 0.575 (0.105) | 0.318 (0.042) | 24.08 (1.602) | 0.511 (0.106) | 22.362 (10.733) |
| TV | 27.03 (1.635) | 0.518 (0.123) | 5.748 (1.283) | 26.22 (1.578) | 0.509 (0.123) | 32.269 (15.414) | 24.12 (1.900) | 0.432 (1.112) | 5.087 (1.049) | 23.70 (1.857) | 0.427 (0.112) | 23.048 (10.854) |
| **End-to-end** | | | | | | | | | | | | |
| Residual UNet | 32.27 (1.810) | 0.808 (0.080) | – | 31.70 (1.970) | 0.785 (0.095) | – | 29.75 (1.675) | 0.750 (0.088) | – | 29.36 (1.746) | 0.733 (0.100) | – |
| E2E-VarNet | 33.40 (2.097) | 0.836 (0.079) | – | 31.71 (2.540) | 0.756 (0.102) | – | 30.67 (1.761) | 0.769 (0.085) | – | 30.45 (1.940) | 0.736 (0.103) | – |
| **PnP diffusion prior** | | | | | | | | | | | | |
| CSGM | 28.78 (6.173) | 0.710 (0.147) | 1.518 (0.433) | 25.17 (6.246) | 0.582 (0.167) | 31.642 (15.382) | 26.15 (6.383) | 0.625 (0.158) | 1.142 (1.078) | 21.17 (8.314) | 0.425 (0.192) | 22.088 (10.740) |
| ScoreMRI | 25.97 (1.681) | 0.468 (0.087) | 10.828 (1.731) | 25.60 (1.618) | 0.463 (0.086) | 33.697 (15.209) | 25.20 (1.526) | 0.405 (0.079) | 8.360 (1.381) | 24.74 (1.481) | 0.403 (0.080) | 24.028 (10.663) |
| RED-diff | 29.36 (7.710) | 0.733 (0.131) | 0.509 (0.077) | 28.71 (2.755) | 0.626 (0.126) | 31.591 (15.368) | 26.76 (6.969) | 0.647 (0.124) | 0.485 (0.068) | 27.33 (2.441) | 0.563 (0.117) | 22.336 (10.838) |
| DiffPIR | 28.31 (1.598) | 0.632 (0.107) | 10.545 (2.466) | 27.60 (1.470) | 0.624 (0.111) | 34.015 (15.522) | 26.78 (1.556) | 0.588 (0.113) | 7.787 (1.741) | 26.26 (1.458) | 0.590 (0.113) | 24.208 (10.922) |
| DPS | 26.13 (4.247) | 0.620 (0.105) | 9.092 (2.925) | 25.83 (2.197) | 0.548 (0.116) | 35.009 (15.967) | 22.82 (4.777) | 0.536 (0.111) | 6.737 (1.928) | 23.00 (3.205) | 0.507 (0.109) | 24.842 (11.263) |
| DAPS | 31.48 (1.988) | 0.762 (0.089) | 1.566 (0.390) | 28.61 (2.197) | 0.689 (0.102) | 31.115 (15.497) | 29.01 (1.712) | 0.681 (0.098) | 1.280 (0.301) | 27.10 (2.034) | 0.629 (0.107) | 22.729 (10.926) |
| PnP-DM | 31.80 (3.473) | 0.780 (0.096) | 4.701 (0.675) | 27.62 (3.425) | 0.679 (0.117) | 32.261 (15.169) | 29.33 (3.081) | 0.704 (0.105) | 3.421 (0.504) | 25.28 (3.102) | 0.607 (0.117) | 22.879 (10.712) |
| LMAPS | **32.83** (2.581) | 0.740 (0.117) | 3.500 (0.544) | **28.77** (1.813) | 0.656 (0.102) | 32.476 (15.303) | **30.50** (2.181) | 0.660 (0.116) | 2.565 (0.399) | **27.43** (1.689) | 0.600 (0.109) | 23.021 (10.804) |

Table 7: Results on black hole imaging. PSNR and Chi-squared of different algorithms on black hole imaging. Gain and phase noise and thermal noise are added based on EHT library.

| Methods | 3% | | | | 10% | | | | 100% | | | |
|---|---|---|---|---|---|---|---|---|---|---|---|---|
| | PSNR | Blur PSNR | $\bar{\chi}^2_{cp}$ | $\bar{\chi}^2_{logca}$ | PSNR | Blur PSNR | $\bar{\chi}^2_{cp}$ | $\bar{\chi}^2_{logca}$ | PSNR | Blur PSNR | $\bar{\chi}^2_{cp}$ | $\bar{\chi}^2_{logca}$ |
| **Traditional** | | | | | | | | | | | | |
| SMILI | 18.51 (1.39) | 23.08 (2.12) | 1.478 (0.428) | 4.348 (3.827) | 20.85 (2.90) | 25.24 (3.86) | 1.209 (0.169) | 21.788 (12.491) | 22.67 (3.13) | 27.79 (4.02) | 1.878 (0.952) | 17.612 (10.299) |
| EHT-Imaging | 21.72 (3.39) | 25.66 (5.04) | 1.507 (0.485) | 1.695 (0.539) | 22.67 (3.46) | 26.66 (3.93) | 1.166 (0.156) | 1.240 (0.205) | 24.28 (3.63) | 28.57 (4.52) | 1.251 (0.250) | 1.259 (0.316) |
| **PnP diffusion prior** | | | | | | | | | | | | |
| DPS | 24.20 (3.72) | 30.83 (5.58) | 8.024 (24.336) | 5.007 (5.750) | 24.36 (3.72) | 30.79 (5.75) | 13.052 (43.087) | 6.614 (26.789) | 25.86 (3.90) | 32.94 (6.19) | 8.759 (37.784) | 5.456 (24.185) |
| LGD | 22.51 (3.76) | 28.50 (5.49) | 15.825 (16.838) | 12.862 (12.663) | 22.08 (3.75) | 27.48 (5.09) | 10.775 (21.684) | 13.375 (56.397) | 21.22 (3.64) | 26.06 (4.98) | 13.239 (17.231) | 13.233 (39.107) |
| RED-diff | 20.74 (2.62) | 26.10 (3.35) | 6.713 (6.925) | 9.128 (19.052) | 22.53 (3.02) | 27.67 (4.53) | 2.488 (2.925) | 4.916 (13.221) | 23.77 (4.13) | 29.13 (6.22) | 1.853 (0.938) | 2.050 (2.361) |
| PnPDM | 24.25 (3.45) | 30.49 (4.93) | 2.201 (1.352) | 1.668 (0.551) | 24.57 (3.47) | 30.80 (5.22) | 1.433 (0.417) | 1.336 (0.478) | 26.07 (3.70) | 32.88 (6.02) | 1.311 (0.195) | 1.199 (0.221) |
| DAPS | 23.54 (3.28) | 29.48 (4.88) | 3.647 (3.287) | 2.329 (1.354) | 23.99 (3.56) | 30.16 (5.13) | 1.545 (0.705) | 2.253 (9.903) | 25.60 (3.64) | 32.78 (5.68) | 1.300 (0.324) | 1.229 (0.532) |
| DiffPIR | 24.12 (3.25) | 30.45 (4.88) | 14.085 (14.105) | 10.545 (8.860) | 23.84 (3.59) | 30.04 (5.03) | 5.374 (3.733) | 5.205 (5.556) | 25.01 (4.64) | 31.86 (6.56) | 3.271 (1.623) | 2.970 (1.202) |
| LMAPS | **24.66** (4.02) | 29.94 (5.17) | **1.497** (0.394) | 4.695 (1.420) | **24.84** (3.695) | 29.98 (5.144) | 1.671 (0.521) | 4.460 (1.555) | **26.79** (3.78) | 32.95 (5.41) | 1.512 (0.474) | 4.622 (1.455) |

Table 8: Comparison between analytical solution and gradient descent for solving LMAPS, LMAPS-GD represents solving LMAPS with gradient descent, LMAPS-A referes to solving LMAPS with analytical solution.

| Task | Method | FFHQ | | | ImageNet | | |
|---|---|---|---|---|---|---|---|
| | | PSNR ↑ | SSIM ↑ | LPIPS ↓ | PSNR ↑ | SSIM ↑ | LPIPS ↓ |
| SR 4× | LMAPS-GD | 30.74 | 0.869 | 0.165 | 26.72 | 0.739 | 0.242 |
| | LMAPS-A | 30.31 | 0.860 | 0.161 | 26.39 | 0.723 | 0.252 |
| Inpaint (Box) | LMAPS-GD | 25.17 | 0.876 | 0.108 | 21.25 | 0.803 | 0.204 |
| | LMAPS-A | 25.35 | 0.871 | 0.120 | 21.15 | 0.796 | 0.216 |

Table 9: Solving Nonlinear Deblurring with different annealing step and gradient step.

| Annealing steps | Gradient steps | FFHQ | | | ImageNet | | |
|---|---|---|---|---|---|---|---|
| | | PSNR ↑ | SSIM ↑ | LPIPS ↓ | PSNR ↑ | SSIM ↑ | LPIPS ↓ |
| 200 | 200 | $29.93_{\pm 1.83}$ | $\mathbf{0.855}_{\pm 0.035}$ | $\mathbf{0.150}_{\pm 0.034}$ | $28.03_{\pm 3.62}$ | $0.774_{\pm 0.099}$ | $0.183_{\pm 0.065}$ |
| 100 | 20 | $27.58_{\pm 1.878}$ | $0.814_{\pm 0.024}$ | $0.200_{\pm 0.040}$ | $26.15_{\pm 3.24}$ | $0.729_{\pm 0.118}$ | $0.257_{\pm 0.079}$ |

# F  LICENSES

**ImageNet Dataset.** The ImageNet dataset is used under the terms of its academic research license. Access requires agreement to ImageNet's data use policy, and redistribution is not permitted. More information is available at: `https://image-net.org/download`.

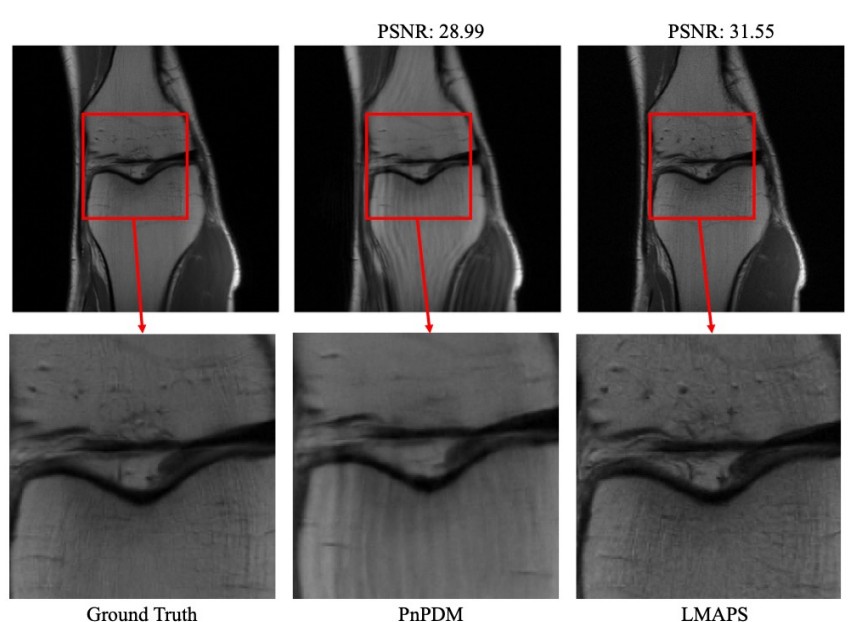

Figure 5: Visualization of CS-MRI restoration (4× raw).

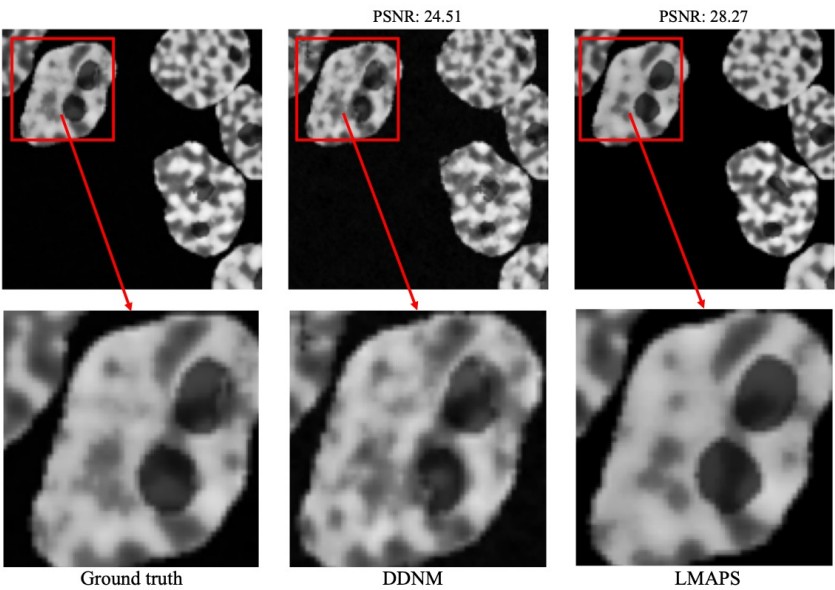

Figure 6: Visualization of Linear Inverse Scattering (Number of receivers = 60).

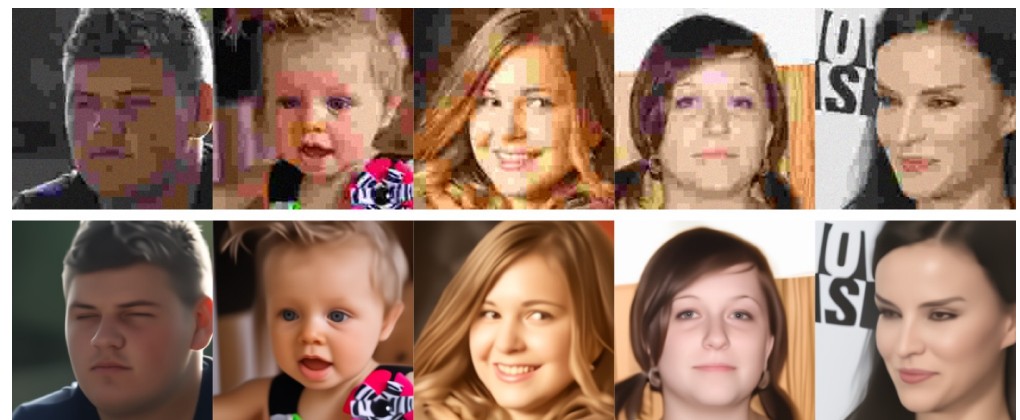

Figure 7: Visualization for solving JPEG restoration (QF=5, $\sigma_y = 0.05$). Top: degraded images; bottom: generated images.

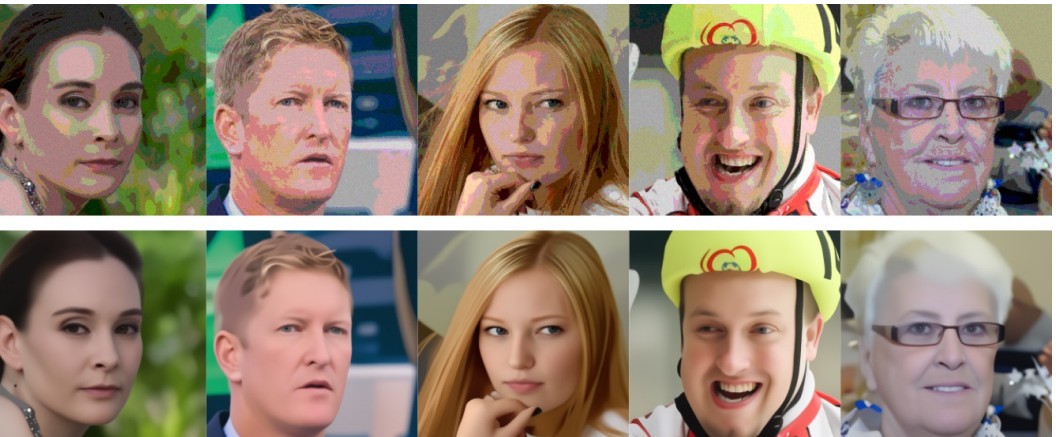

Figure 8: Visualization for solving Quantization (2 bit). Top: degraded images; bottom: generated images.

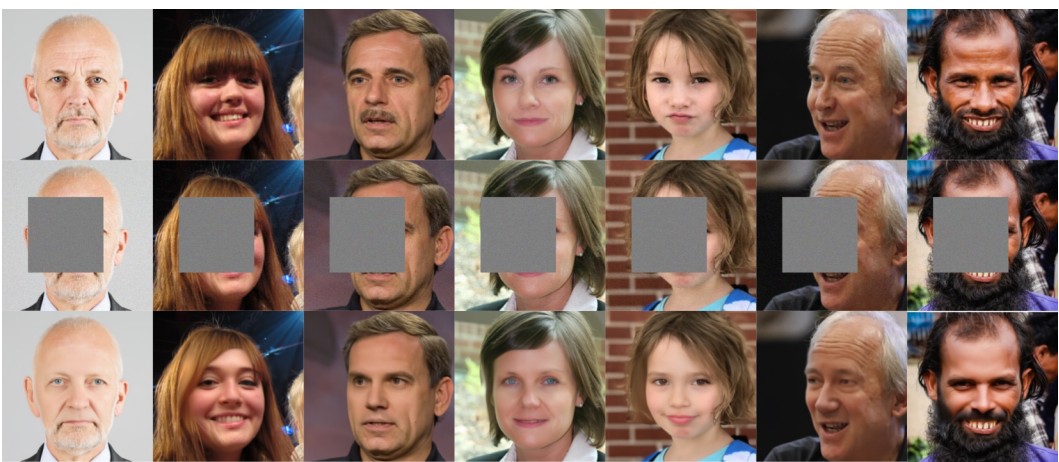

Figure 9: Visualization for solving Inpaint (Box). Top: ground truth; middle: degraded images; bottom: generated images.

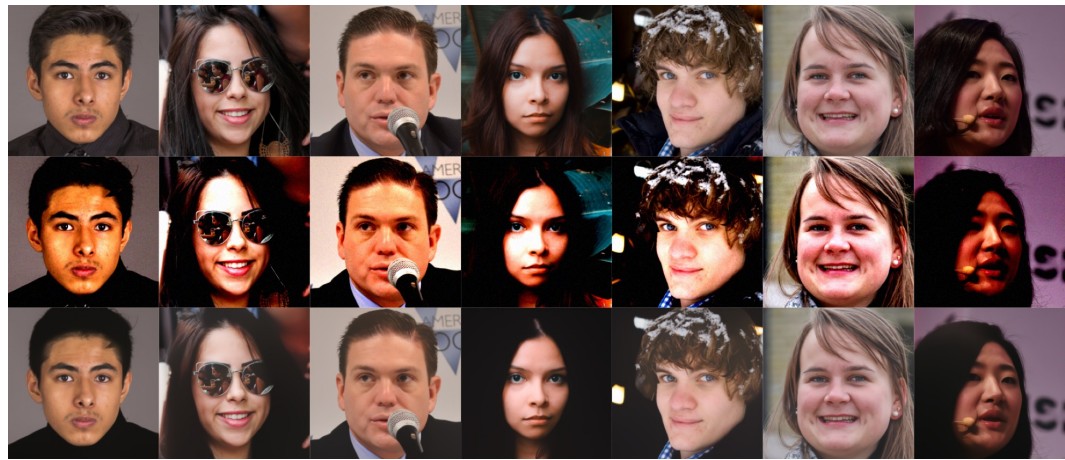

Figure 10: Visualization for solving HDR. Top: ground truth; middle: degraded images; bottom: generated images.

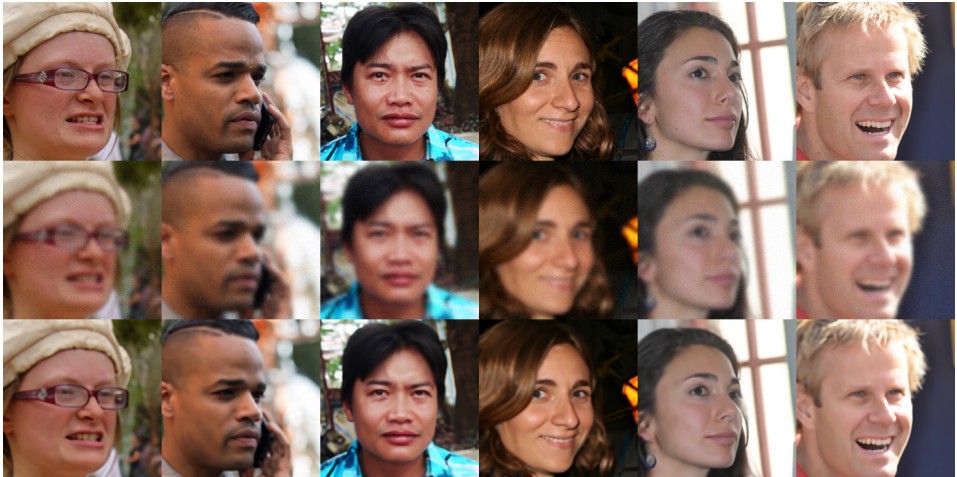

Figure 11: Visualization for solving Deblurring. Top: ground truth; middle: degraded images; bottom: generated images.

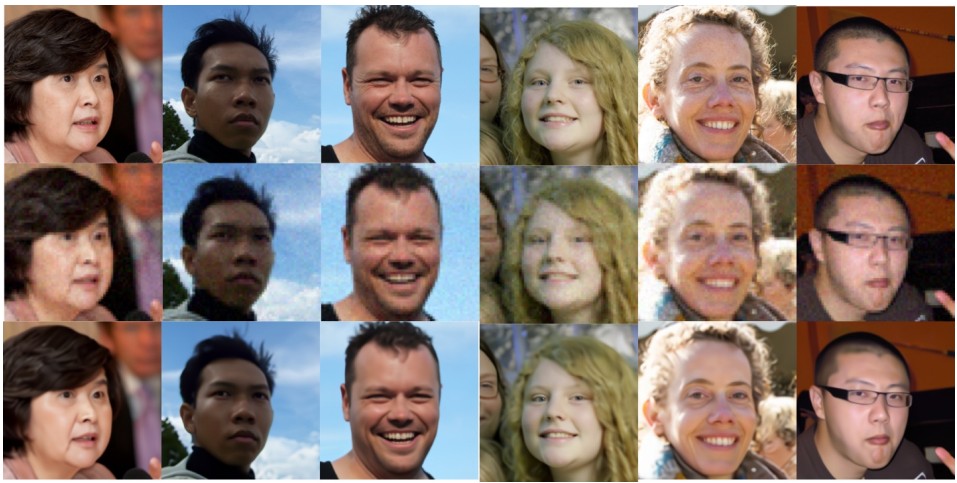

Figure 12: Visualization for solving Super-Resolution. Top: ground truth; middle: degraded images; bottom: generated images.

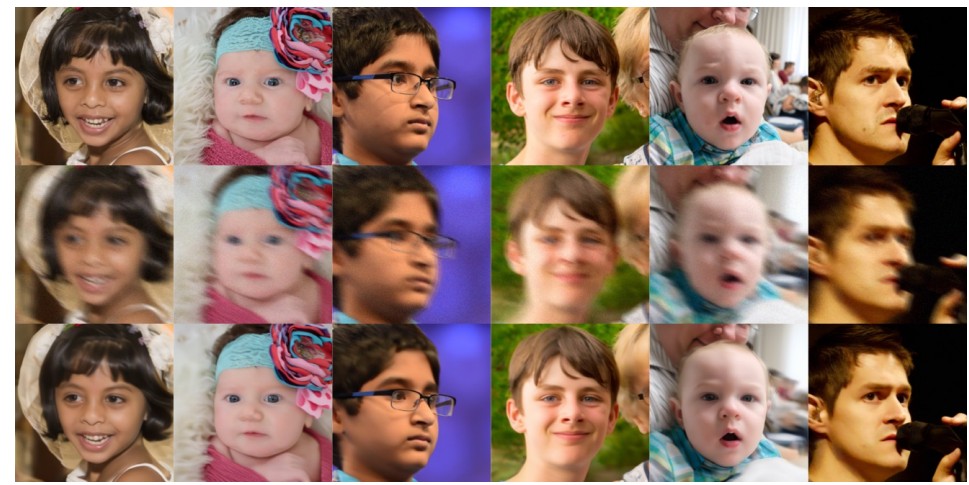

Figure 13: Visualization for solving Nonlinear Deblurring. Top: ground truth; middle: degraded images; bottom: generated images.

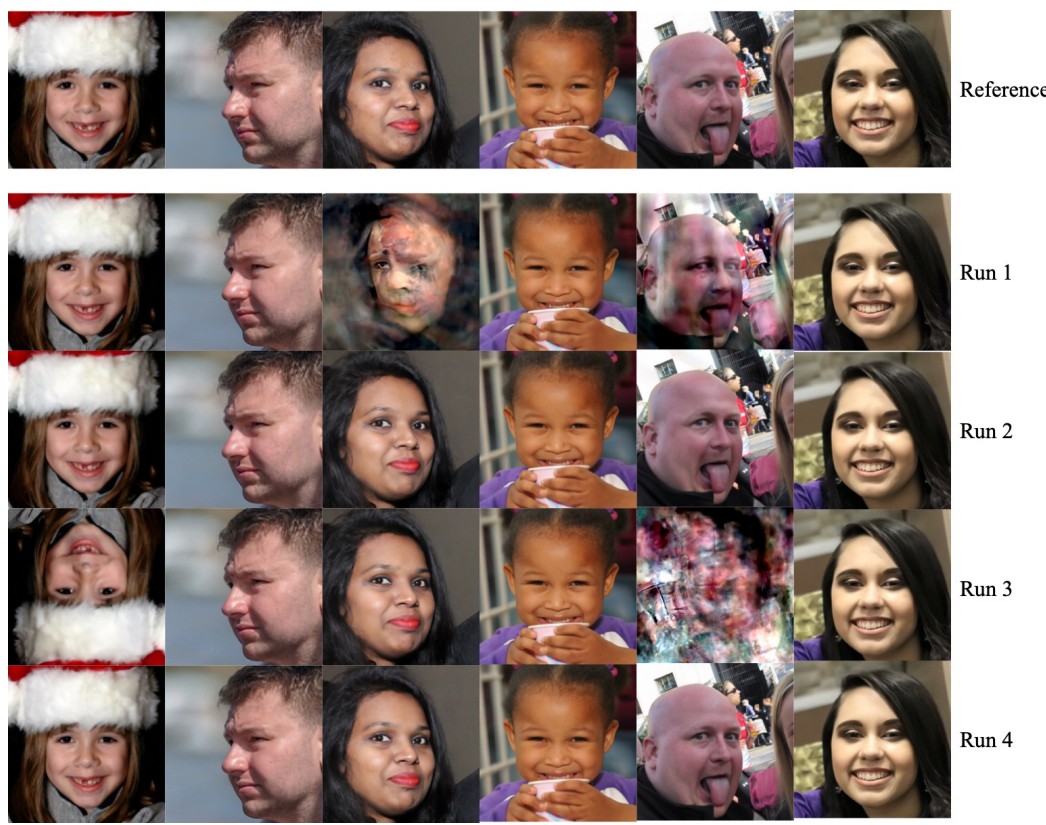

Figure 14: Visualization for solving Phase retrieval.

