# OpenReview forum: "Local MAP Sampling for Diffusion Models"
_ICLR.cc/2026/Conference — Submitted to ICLR 2026_

### Official Review · Reviewer_Bng9 · 2025-10-27

**Soundness:** 3
**Presentation:** 3
**Contribution:** 2
**Rating:** 4
**Confidence:** 5

**Summary:**

This paper develops an inference time only algorithm for solving inverse problems while leveraging a pre-trained diffusion models. Its main contribution is the formalization of a local maximum a posterior framework for this purpose; at each step of the diffusion process, one computes the maximum a posterior of the clean state distribution conditioned on the observation and the current diffusion state, before sampling back a new noisy state.

**Strengths:**

- The paper is pretty straightforward to read and easy to understand.
- It tries to reframe various existing methods within a single framework, which is commendable. For example, the paper provides a reinterpretation of TMPD which I find to be actually more interesting than the explanation provided in the original paper [1]. Indeed, in the original paper the authors say that they justify their method is a a new guidance paper but the stopgrad operation used breaks this interpretation. The local MAP framework here is a much sounder explanation that actually recovers what TMPD implements and explains why the stopgrad operation is not so bad after all.
- Various empirical choices are made and these seem to improve the performance with respect to most existing baselines.

[1] Boys, B., Girolami, M., Pidstrigach, J., Reich, S., Mosca, A. and Akyildiz, O.D., 2023. Tweedie moment projected diffusions for inverse problems.

**Weaknesses:**

A major weakness of the paper is that it makes various imprecise/false claims and does not describe existing methods accurately:
- In Section 3.2 for example, line 185 to 187, the authors basically claim that DPS and their framework coincides when one uses a "Gaussian diffusion prior approximation" (what does this mean exactly?) but this is not explained in the next section and I don't see how any special cases of 11 or 12 yield DPS.
- In Figure 1 we can see that PiGDM is said to be a special case of LMAPS but here again I am not sure that this is the case. Can the authors clarify for which choice of covariance this holds?
- The description of DAPS is also inaccurate; DAPS advocates the use of the very specific choice of $\rho_t = 1$, which ensures the "decoupling". Hence (6) is not really what DAPS implements.

The claims of the paper are also somewhat exaggerated:
- In line 73  "replacing heuristic choices in existing solvers." but the paper actually proposes a heuristic approach not really backed with theory. For example, is the upperbound $\Sigma_{0|t} \leq (\sigma^2 _t / \alpha^2 _t) I$ actually true? The authors should either remove this sentence of explicitely claim that all the design choices made in the paper are heuristic (which is not a critique). Furthermore, the various heuristic choices that are made seem to be very specific to a range of NFEs, since it seems that increasing the NFEs does not necessarily improve the performance.
- line 74 "develop a gradient approximation strategy for non-differentiable operators", here this is not really a contribution, as the gradient approximation is actually quite naive. For example there are already differentiable versions of JPEG which are much less naive that what the paper proposes [2].

Regarding the numerical experiments, they are exhaustive but I regret the comparison with more recent methods which outperform many of the baselines included, for example [3, 4]

Overall, the paper is interesting but I feel that fails to fully achieve its stated objectives.

[2] Reich, C., Debnath, B., Patel, D. and Chakradhar, S., 2024. Differentiable jpeg: The devil is in the details.
[3] Rozet, F., Andry, G., Lanusse, F. and Louppe, G., 2024. Learning diffusion priors from observations by expectation maximization.
[4] Janati, Y., Moufad, B., Abou El Qassime, M., Durmus, A.O., Moulines, E. and Olsson, J., 2025, February. A Mixture-Based Framework for Guiding Diffusion Models.

**Questions:**

For linear inverse problems, does using the exact solution of (13) yield better performance than gradient descent?

---

> ### Author Response · Authors · 2025-11-21
>
> # Part 1
>
> ## Response to Weaknesses 1
>
> > In Section 3.2 for example, line 185 to 187, the authors basically claim that DPS and their framework coincides when one uses a "Gaussian diffusion prior approximation" (what does this mean exactly?) but this is not explained in the next section and I don't see how any special cases of 11 or 12 yield DPS.
>
> In section 4.2, we explained that LMAPS reduced to Tweedie Moment-Projected Diffusion, which is the exact solution of DPS for linear inverse problem under a Gaussian diffusion prior approximation: $p(x_0 \mid x_t) = \mathcal{N} (x_0; m_{0\mid t}, \Sigma_{0\mid t})$. To avoid confusion, we have now **explicitly added the derivation** in Section 4.2 showing how LMAPS reduces to the Tweedie Moment-Projected Diffusion (TMPD), which is the _exact_ form of DPS for linear inverse problems under this approximation. Specifically, for linear inverse problems $p(y \mid x_0) = \mathcal{N} (H x_0, \sigma_y^2 \mathbb{I} )$ with a Gaussian assumption on the prior $p(x_0 \mid x_t) = \mathcal{N} (x_0; m_{0\mid t}, \Sigma_{0\mid t})$.   Solving the local MAP in closed form, we have
>
> $$
>     x_0^* = m_{0 \mid t} + \Sigma_{0 \mid t} H^T (H \Sigma_{0 \mid t} H^T + \sigma_y^2 \mathbb{I})^{-1} (y - H m_{0\mid t})
>      = m_{0 \mid t} + \frac{\sigma_t^2}{\alpha_t} \nabla_{x_t} \log p(y \mid x_t) $$
>
> > In Figure 1 we can see that PiGDM is said to be a special case of LMAPS but here again I am not sure that this is the case. Can the authors clarify for which choice of covariance this holds?
>
> Thank you for pointing out this inaccuracy, we have removed this claim from the Figure.   Our proposed method, later results, and experiments do not rely on this relationship.
>
> > The description of DAPS is also inaccurate; DAPS advocates the use of the very specific choice of $\rho_t=1$, which ensures the "decoupling". Hence (6) is not really what DAPS implements.
>
> Thanks. We have revised this step to be consistent with the original DAPS formulation, and we have also confirmed that our implementation for the experiment matches the original DAPS.
>
> ## Response to Weaknesses 2
>
> > The claims of the paper are also somewhat exaggerated:
> In line 73 "replacing heuristic choices in existing solvers." but the paper actually proposes a heuristic approach not really backed with theory. For example, is the upperbound $\Sigma_{0\mid t} \leq (\sigma_t^2/\alpha_t^2) I$ actually true? The authors should either remove this sentence of explicitly claim that all the design choices made in the paper are heuristic (which is not a critique).
>
>
> We have revised the wording around “replacing heuristic choices” to avoid over-claiming: in the updated version, we now argue that covariance approximation is motivated by Gaussian prior assumption, rather than a fully exact characterization, and we explicitly acknowledge that some design choices remain heuristic in practice.
>
> Regarding the bound $\Sigma_{0 \mid t} \preceq \frac{\sigma_t^2}{\alpha_t^2} \mathbb{I}$, this is indeed holds in the linear–Gaussian setting. For a Gaussian prior $x_0 \sim \mathcal{N}(\mu_0, \Sigma_0)$, the exact posterior covariance under the forward noising process $x_t = \alpha_t x_0 + \sigma_t \epsilon$, $\epsilon \sim \mathcal{N}(0,\mathbb{I})$ is
>
> $$\Sigma_{0 \mid t}
>     = \bigl(\Sigma_0^{-1} + \tfrac{\alpha_t^2}{\sigma_t^2} \mathbb{I}\bigr)^{-1}
>     = \frac{\sigma_t^2}{\alpha_t^2} \mathbb{I}
>       + \mathcal{O}\left( \bigl(\tfrac{\sigma_t^2}{\alpha_t^2}\bigr)^2 \right)
>     \preceq \frac{\sigma_t^2}{\alpha_t^2} \mathbb{I},$$
> More generally, even for non-Gaussian priors $p(x_0)$ with a smooth log-density, the local Gaussian approximation to $p(x_0 \mid x_t)$ is asymptotically isotropic. We provide detailed proof in the Appendix B.
>
>
>
> [2] Reich, C., Debnath, B., Patel, D. and Chakradhar, S., 2024. Differentiable jpeg: The devil is in the details.
>
> [3] Rozet, F., Andry, G., Lanusse, F. and Louppe, G., 2024. Learning diffusion priors from observations by expectation maximization.
>
> [4] Janati, Y., Moufad, B., Abou El Qassime, M., Durmus, A.O., Moulines, E. and Olsson, J., 2025, February. A Mixture-Based Framework for Guiding Diffusion Models.
>
> [5] Gutha et al. Inverse Problems with Diffusion Models: A MAP Estimation Perspective.
>
> [6]Wang, Yinhuai, Jiwen Yu, and Jian Zhang. "Zero-shot image restoration using denoising diffusion null-space model." arXiv preprint arXiv:2212.00490 (2022).
>
> [7] Song J, Vahdat A, Mardani M, et al. Pseudoinverse-guided diffusion models for inverse problems[C]//International Conference on Learning Representations. 2023.

---

> > ### Author Response · Authors · 2025-11-21
> >
> > # Part 2
> >
> > > Furthermore, the various heuristic choices that are made seem to be very specific to a range of NFEs, since it seems that increasing the NFEs does not necessarily improve the performance.
> >
> >
> >
> > We agree that for several cases, increasing NFEs does not always mean better performance. For instance, in our ablation study on optimization steps vs. diffusion steps (NFEs) for Gaussian Deblurring in Figure 4, the best LPIPS is achieved by NFE=50. But we argue that the reasons behind this are multifaceted. For instance, for a fixed measurement model and noise level, the inverse problem itself admits a performance ceiling: once the solution has converged to a region consistent with the measurements and the learned prior, additional function evaluations mainly refine already small residual errors or stochastic fluctuations, which may not translate into improved perceptual or distortion metrics.
> >
> > > line 74 "develop a gradient approximation strategy for non-differentiable operators", here this is not really a contribution, as the gradient approximation is actually quite naive. For example there are already differentiable versions of JPEG which are much less naive that what the paper proposes [2].
> >
> > We removed the broad claim about general non-differentiable operators and only kept the JPEG example.
> >
> >
> > ## Response to Weaknesses 3
> >
> > > Regarding the numerical experiments, they are exhaustive but I regret the comparison with more recent methods which outperform many of the baselines included, for example [3, 4]
> >
> > We thank the reviewer for the suggestion. Our experiments focus on a representative subset of widely adopted diffusion-based inverse problem methods to ensure clear and fair comparisons. Incorporating every recent method is challenging due to feasibility constraints, but we agree that including the most relevant ones strengthens the evaluation.
> >
> >
> > In the revised version, we have added **MGDM** [4], **MMPS** [3]. Furthermore, we have added the most recent MAP solver method **MAP-GA** [5] in Table 1. After updating the experiments, LMAPS still achieves the best performance in 43 out of 60 cases. Compared to MAP-GA and DMPlug, LMAPS achieves better performance in 17 out of 18 cases.
> >
> >
> > ## Response to Question 1
> >
> > > For linear inverse problems, does using the exact solution of (13) yield better performance than gradient descent?
> >
> > We have added a direct comparison between solving $x_0^*$ using gradient-based optimization and the closed-form solution for SR $8\times$ and Inpainting (Box) in Table 8 and Appendix E.3. The results demonstrate that the performance of the two approaches is highly comparable. In particular, the PSNR difference is consistently below 0.5 dB, and the LPIPS difference is less than 0.015 across both FFHQ and ImageNet for these two linear inverse problems.
> >
> > These findings indicate that the closed-form solution achieves nearly identical reconstruction quality to gradient-based optimization while being more efficient. Therefore, when the forward operator admits an SVD, we recommend using the closed-form solution in practice.

---

### Official Review · Reviewer_nKLJ · 2025-10-31

**Soundness:** 2
**Presentation:** 2
**Contribution:** 2
**Rating:** 4
**Confidence:** 4

**Summary:**

The paper proposes Local MAP sampling, which is a plug-and-play inference framework for solving inverse problems with diffusion models. The idea is to iteratively solve local MAP subproblems along the reverse diffusion process.  This framework also consolidates existing optimization-based approaches under a common probabilistic perspective. The proposed algorithm (LMAPS) outperforms existing approaches on a wide range of imaging and scientific inverse problems.

**Strengths:**

The paper consolidates existing optimization-based diffusion inverse solvers under a common probabilistic perspective, providing new insights into the effectiveness of the methods.

The empirical validation is quite impressive with extensive experiments across a wide range of scientific and imaging inverse problems, and comparison against existing methods. This clearly indicates the empirical effectiveness of the proposed algorithm (LMAPS).

The paper is generally well-written and easy to read. The relationship with other existing methods is clearly articulated, which makes the probabilistic perspective and the similarities and distinctions of LMAPS with other optimization-based works easier.

**Weaknesses:**

While the paper provides an insightful perspective on posing inverse problem solving as local MAP sampling, it also has several concerns (as listed) that need to be addressed. I would be willing to increase the score if the following concerns can be addressed sufficiently.

Q1. While the proposed local MAP sampling framework is clearly empirically effective, I generally don’t find the theoretical formulation of the local MAP objective very novel, as the main insight is heavily based on, and can be trivially observed from the previous work DAPS, as follows.

Since LMAPS changes the sampling procedure $x^{daps}_0 \sim p(x_0|x_t,y)$ in DAPS to the optimization procedure $x^{lmaps}_0 = \arg\max p(x_0|x_t,y)$, it is apparent that $x^{lmaps}_0$ will generally be a higher likelihood sample of $p(x_0|x_t,y)$ compared to $x^{daps}_0$, which in the end, can result in a highly likely solution $x_0$ in the case of LMAPS when compared to DAPS. I’d appreciate the authors’ comments in case I’m missing something regarding this main contribution.


Q2. From Q1, and also from Sec 3.2, in general settings, one can intuitively expect LMAPS to not necessarily find a global MAP solution or follow exact posterior sampling but rather tend to give highly likely posterior samples (as Line189 essentially implies). However, I believe a theoretical analysis (maybe in a toy setting as in Appendix A), showing the extent to which the samples of the LMAPS are biased towards highly-likely regions of the posterior, can immensely strengthen the paper’s contribution.


Q3. Eq 14 can arise out of using proximal schemes such as HQS or ADMM, etc, to solve the original MAP problem itself (See [1]). In this regard, the claim about other optimization-based methods not being probabilistically interpretable may not be completely true. Also, I’m quite unsure about the fairness of the claim regarding “the local MAP sampling framework providing a probabilistic interpretation to optimization-based methods”, as it seems heavily based on a Gaussian assumption of $p(x_0|x_t)$ (Lines 264-289). Can the authors clarify?

Q4. The majority of the compared baselines seem to be posterior sampling methods, while LMAPS, for a fair comparison, needs to consider recent MAP solver methods such as [2] and [3] (which was mentioned as being considered in Line 360, but wasn’t actually considered for experiments)

Q5. The linear inverse problem experiments (i.e., box, random inpainting) are typically considered easier tasks compared to tasks like large-hole inpainting or cases where there are only a few measurements. Such tasks can highlight significant differences in metrics among different methods and can suggest the true effectiveness of the method. Also, note that for image restoration, high PSNR doesn’t necessarily correspond with high perceptual quality, unlike LPIPS, which is a more favorable and important metric of choice (See [4]).

Q6. In Table 3. Does “The non-parallel single-image sampling time” mentioned in the caption mean that each method is run with a batch size of 1, as it is supposed to be?  Also, since this task is for deblurring, as mentioned in point 5, it is appropriate to report LPIPS instead of PSNR.

Q7. FFHQ is also typically considered an easier dataset than Imagenet. Some qualitative examples of Imagenet would provide more insight into the true effectiveness of LMAPS compared to other methods.

[1] Zhu et al. Denoising Diffusion Models for Plug-and-Play Image Restoration
[2] Gutha et al. Inverse Problems with Diffusion Models: A MAP Estimation Perspective
[3] Wang et al. DMPlug: A Plug-in Method for Solving Inverse Problems with Diffusion Models
[4] Blau et al. The Perception-Distortion Tradeoff

**Questions:**

Please see the questions above.

---

> ### Author Response · Authors · 2025-11-21
>
> ## Response to Q1
>
> More accurately, LMAPS replace $\mathbb{E}[x_0 \mid x_t, y]$ of DPS by $x_0^*=\arg \max p(x_0 \mid x_t, y)$, because in Eq. (6) $x_{t-\Delta t}\sim \mathcal{N}(\alpha_{t- \Delta t} x_0, \sigma_{t-\Delta t}^2 \mathbb{I})$ is different from DPS and LMAPS. The goal of DPS and DAPS are to draw samples from $p(x_0 \mid y)$, which is fundamentally different from LMAPS.
> Indeed, our goal is to prioritize high likelihood samples over sampling the full posterior.
>
> To recount our main contributions, LMAPS provides a  theoretical framework connecting DPS, TMPD, and optimization-based solvers through **local MAP subproblems along the diffusion trajectory**, revealing behavior fundamentally different from DPS, DAPS, or global MAP. LMAPS also provides new insight on TMPD and optimization based methods. For instance, as Reviewer Bng9 suggested, LMAPS provides a reinterpretation of TMPD which is “more interesting than the explanation provided in the original paper” and yields insight into implementation decisions in TMPD.   Moreover, LMAPS introduces new tools (covariance approximation, objective reformulation) for making local MAP practically effective across diverse inverse problems.
>
> ## Response to Q2
>
> In the revised Appendix A, we added a concrete theoretical analysis for a Gaussian mixture prior, where both the posterior and the local MAP estimator admit closed-form expressions. LMAPS preferentially follows posterior modes with large posterior mass while avoiding low-density or between-mode regions. This theoretical behavior is fully consistent with our empirical observations. As illustrated in Figure 3, LMAPS samples concentrate around dominant posterior modes, whereas regions of low posterior density receive almost no mass—unlike posterior mean–based updates or DPS sampling.
>
> ## Response to Q3
>
> We agree with the reviewer that proximal methods such as HQS or ADMM have a well-known MAP interpretation when applied directly to the global MAP objective. To avoid confusion, we have removed the earlier claim suggesting otherwise. Our contribution is not to reinterpret HQS/ADMM themselves, but to show that **optimization-based diffusion solvers**—whose updates operate along the diffusion trajectory—can be viewed as solving **local MAP subproblems** with respect to the diffusion posterior $p(x_0 \mid x_t, y)$. This probabilistic connection to diffusion-based inference was not formalized in prior work.
>
> We agree that our probabilistic interpretations rely on the assumption that $p(x_0\mid x_t)$ is a Gaussian with isotropic variance. This Gaussian assumption has been widely used in DPS methods, such as $\Pi$GDM [6]. For non-isotropic Gaussian, the Local MAP objective becomes:
>
> $$x_0^*
>     = \arg \min (x_0 - m_{0 \mid t})^{T} \Sigma_{0 \mid t}^{-1} (x_0 - m_{0 \mid t}) + \frac{1}{\sigma_y^2} \Vert y - \mathcal{H} (x_0) \Vert^2.$$
>
> This objective generalizes previous optimization-based methods by allowing a  weighted matrix $\Sigma_{0\mid t}$ to align with $m_{0 \mid t}$. This view provides a new interpretation for TMPD.
>
> ## Response to Q4
>
> > The majority of the compared baselines seem to be posterior sampling methods
>
> We included the most recent optimization based method - SITCOM [5] as a baseline, which demonstrates better performance than DMPlug [3]  on most inverse problems.
>
> > needs to consider recent MAP solver methods such as [2] and [3]
>
> Thank you for this suggestion, we have added MAP-GA [2] and DMPlug [3] in Table 1. After updating the experiments, LMAPS still achieves the best performance in 43 out of 60 cases. Compared to MAP-GA and DMPlug, LMAPS achieves better performance in 17 out of 18 cases.
>
> ## Response to Q5
>
> We appreciate the reviewer’s suggestion. For image restoration tasks, our evaluation already includes LPIPS. For other inverse problems, we follow the standard baselines and metrics used in prior literature.
>
> In addition, our evaluation covers a broad range of scientific inverse problems, including CS-MRI, linear inverse scattering, and black hole imaging, providing a comprehensive assessment of LMAPS across diverse and challenging settings.
>
> ## Response to Q6
>
> Yes, “The non-parallel single-image sampling time” mentioned in the caption means that each method is run with a batch size of 1. As the reviewer suggested, we updated the table to report LPIPS instead of PSNR.
>
> ## Response to Q7
>
> In Table 1, we have included ImageNet as qualitative examples.
>
>
>
> [1] Zhu et al. Denoising Diffusion Models for Plug-and-Play Image Restoration [2] Gutha et al. Inverse Problems with Diffusion Models: A MAP Estimation Perspective [3] Wang et al. DMPlug: A Plug-in Method for Solving Inverse Problems with Diffusion Models [4] Blau et al. The Perception-Distortion Tradeoff [5] Alkhouri et al. Sitcom: Step-wise triple-consistent diffusion sampling for inverse problems [6] Song et al. Pseudoinverse-guided diffusion models for inverse problems

---

### Official Review · Reviewer_8LcP · 2025-10-31

**Soundness:** 1
**Presentation:** 2
**Contribution:** 1
**Rating:** 2
**Confidence:** 5

**Summary:**

The authors propose a framework for addressing inverse problems using diffusion priors.
The framework operates by solving a sequence of optimization problems, each of which can be interpreted as a Local-MAP estimation step.
Since these optimization problems depend on the covariance of $X_0 | X_t$, which is computationally expensive to handle directly, the authors motivate a new interpretable approximation strategy.
Extensive benchmarks were conducted to validate the framework.

**Strengths:**

- Attempt to unify optimization based-method into a one framework
- rethink the approximation of the covariance $X_0 | X_t$ involved in the local-MAP problem

**Weaknesses:**

**Paper exposition**

The central idea of the paper on inverse problems and that the goal is MAP estimation is a problematic premise.
In inverse problems, the degradation operator induces information loss (low-rank component plus noise) so multiple distinct solutions can fit the observations equally well; representing that multiplicity is a feature, not a flaw.
Reducing solving inverse problem to MAP eliminate a fundamental part about capturing the variability of the solutions, uncertainty quantification, critical in several applications.

The exposition of the background on diffusion models is not well-written and might be confusing.
Similarly, it is concerning to see the assertion "there are two main approaches to do posterior sampling with diffusion prior." (Line 102-103) being presented as a categorical truth.
In fact, the literature contains a broad set of posterior-sampling methods and several survey efforts attempts to summaries and unify them; see for instance [1] and [2]
Reducing that diversity into two lines misrepresents the state of the art


**Technical flaws**

- In Algorithm 2: the second step (Line 5) in DAPS is not correct, the paper treats the second step as a single operation, whereas it amounts to simulation of the reverse diffusion process.

- The authors repeatedly treats the maximizer as unique, although the considered step may have many; e.g. Lines 146 and 190–191.
This is incorrect in general: the arg-max of an objective function is a set, and convergence to a particular maximizer depend on initialization (for example, the function $x \mapsto -x^4+8x^2$ has argmax $ \\{ -2, 2 \\} $)

- The approximation of the covariance $X_0 | X_t$ has little novelty. The approximation reduces to a spherical covariance model while introducing two additional hyperparameters k_1 and k_2 fo; this both resembles prior covariance approximations in the literature. The introduced hyperparameters $k_1$ and $k_2$ are treated as independent although they were earlier related to the same quantity $2 k_2/ k_1^2 = k / (\alpha_t^2\sigma_y^2)$.

- the statement (Line 686) about the denoiser is incorrect, it states that "this estimator is mode-averaging, and may fall between mixture modes". Although its expression might give this deceptive impression, the authors disregard the fact that the weights of the components depend on $x_t$


**Typos/mistakes**

- Line 230: it is not the correct expression of the $Cov(X_0 | X_t=x_t)$, it must be $Cov(X_0|X_t) = \sigma_t/\alpha_t \nabla m_{0|t}$; see equation (9) in [3]
- k is introduced in Equation 12, but it is not clear what it refers to
- Line 216-218: deforms what is being said in the [3] [see Proposition 1 in [3]], this actually the definition of the conditional covariance, and has nothing to do with "faithfully reflect the covariance of x0 | xt"


---

... [1] Daras, Giannis, et al. "A survey on diffusion models for inverse problems." arXiv preprint arXiv:2410.00083 (2024).

... [2] Oliviero-Durmus, Alain, et al. "Generative modelling meets Bayesian inference: a new paradigm for inverse problems." Philosophical Transactions A 383.2299 (2025): 20240334.

... [3] Boys, Benjamin, et al. "Tweedie moment projected diffusions for inverse problems." arXiv preprint arXiv:2310.06721 (2023)

**Questions:**

- Can the authors provide clarification on Figure 3, namely the inverse problem setup? it seems the samples are generated from the prior distribution.
- Can the authors provide clarification on claim in Lines 302–305 about using $K$ gradient steps despite a closed-form solution? If the operator admits an SVD (e.g., inpainting), the advantage of iterative updates is unclear.

---

> ### Author Response · Authors · 2025-11-21
>
> # Part 1
>
> ## Response to Weakness 1 - Paper exposition
>
> > The central idea of the paper on inverse problems and that the goal is MAP estimation is a problematic premise
>
> We agree that multiple distinct solutions can fit the observations equally well, but we respectively disagree that the goal of MAP estimation is a problematic premise. Our intention was not to diminish the importance of posterior sampling, but to highlight that MAP estimation has also been a long-standing and widely used objective in classical inverse problems, e.g., imaging, medical, and geophysical applications [4-6]. Accordingly, we have softened the phrasing in the introduction to reflect this balance and to clarify that MAP and posterior sampling serve complementary roles.
>
>
>
> > it is concerning to see the assertion "there are two main approaches to do posterior sampling with diffusion prior." (Line 102-103) being presented as a categorical truth. In fact, the literature contains a broad set of posterior-sampling methods and several survey efforts attempts to summarize and unify them; see for instance [1] and [2] Reducing that diversity into two lines misrepresents the state of the art.
>
> To clarify, our statement referred specifically to diffusion posterior sampling as a class of methods that apply Tweedie’s formula to estimate the posterior mean:
>
> $$    \mathbb{E}[x_0 \mid x_t, y] = m_{0\mid t} + \frac{\sigma_t^2}{\alpha_t} \nabla_{x_t} \log p(y \mid x_t).
> $$
> As discussed in [1], many existing methods—including the variants illustrated in Fig. 1 of [1]—can be viewed as instances or approximations of this Tweedie-based diffusion posterior sampling framework. Our intention was to highlight this unifying structure rather than to reduce the space of prior work.
>
> To avoid misunderstanding, we revised this sentence to be: “In this paper, we focus on two representative lines of posterior sampling approaches with diffusion priors: (i) the family of diffusion posterior sampling (DPS) methods based on Tweedie's formula, and (ii) Decoupled Annealing Posterior Sampling (DAPS).”
>
> ## Response to Weakness 2 - Technical flaws
>
> > In Algorithm 2: the second step (Line 5) in DAPS is not correct, the paper treats the second step as a single operation, whereas it amounts to simulation of the reverse diffusion process.
>
> Thanks. We have revised this step to be consistent with the original DAPS formulation. We would like to clarify that the experimental implementation already followed the correct procedure, and only the algorithm description required correction.
>
> > The authors repeatedly treats the maximizer as unique, although the considered step may have many; e.g. Lines 146 and 190–191. This is incorrect in general: the arg-max of an objective function is a set, and convergence to a particular maximizer depend on initialization (for example, the function $x \mapsto -x^4 + 8x^2$ has argmax $\{-2, 2 \}$
>
> We agree that the argmax of the posterior is in general a set. In the revised manuscript, we avoid wording that implies uniqueness and instead refer to “a maximizer” of the posterior. At the same time, we note that in Bayesian statistics and inverse problems, MAP estimation is conventionally described as a _point estimation_ procedure (e.g. [4], Stuart, 2010, page 459), even though the argmax may contain multiple elements.
>
> [1] Daras, Giannis, et al. "A survey on diffusion models for inverse problems." arXiv preprint arXiv:2410.00083 (2024).
>
> [2] Oliviero-Durmus, Alain, et al. "Generative modelling meets Bayesian inference: a new paradigm for inverse problems." Philosophical Transactions A 383.2299 (2025): 20240334.
>
> [3] Boys, Benjamin, et al. "Tweedie moment projected diffusions for inverse problems." arXiv preprint arXiv:2310.06721 (2023)
>
> [4] Andrew M Stuart. Inverse problems: a bayesian perspective. Acta numerica, 19:451–559, 2010.
>
> [5] Jari P Kaipio and Erkki Somersalo. Statistical and computational inverse problems. Springer, 2005.
>
> [6] Albert Tarantola. Inverse problem theory and methods for model parameter estimation. SIAM, 2005

---

> ### Author Response · Authors · 2025-11-21
>
> # Part 2
>
> > The approximation of the covariance $X_0 \mid X_t$ has little novelty. The approximation reduces to a spherical covariance model while introducing two additional hyperparameters k_1 and k_2 fo; this both resembles prior covariance approximations in the literature.
>
> We agree that spherical (diagonal and isotropic) covariance approximations have appeared in prior diffusion-based inverse problem methods, and our goal is not to claim novelty in the functional form of the covariance itself. Instead, our contribution lies in providing a **probabilistically interpretable derivation** of this approximation from the conditional distribution $X_0 \mid X_t$, clarifying how it arises naturally from the local MAP formulation and how it interfaces with the forward operator. Specifically,
>
> For a Gaussian prior $x_0 \sim \mathcal{N}(\mu_0, \Sigma_0)$, the exact posterior covariance under the forward noising process $x_t = \alpha_t x_0 + \sigma_t \epsilon$, $\epsilon \sim \mathcal{N}(0,\mathbb{I})$ is
>
>    $$  \Sigma_{0 \mid t}
>     = \bigl(\Sigma_0^{-1} + \tfrac{\alpha_t^2}{\sigma_t^2} \mathbb{I}\bigr)^{-1}
>     = \frac{\sigma_t^2}{\alpha_t^2} \mathbb{I}
>       + \mathcal{O} \left( \bigl(\tfrac{\sigma_t^2}{\alpha_t^2}\bigr)^2 \right)
>     \preceq \frac{\sigma_t^2}{\alpha_t^2} \mathbb{I},$$
>
> so the leading term is isotropic and all anisotropy appears only as higher–order corrections as $t \to 0$ (i.e., $\sigma_t^2 \to 0$ and $\alpha_t \to 1$).
>
> More generally, even for non-Gaussian priors $p(x_0)$ with a smooth log-density, local Gaussian approximation to $p(x_0 \mid x_t)$ is asymptotically isotropic. We provide a formal statement and proof in Appendix B.
>
> > The introduced hyperparameters $k_1$ and $k_2$ are treated as independent although they were earlier related to the same quantity $2k_2/k_1^2=k/(\alpha_t^2\sigma_y^2)$.
>
> Treating them as independent does not violate any theoretical constraint, since $k$ is a free parameter. This would allow us to set k1, k2 independently (while still respecting the identity) by varying k.  In other words, choose k1, choose k2, back out the implied k.  Viewing the hyperparameters $k_1$ and $k_2$ in this way provides advantages, as detailed in Sec 4.1.  For instance, keeping the weights $(1 - \frac{\sigma_t^2}{\sigma_t^2 + k_1^2} )$ and $\frac{\sigma_t^2}{\sigma_t^2 + k_1^2} $ in $[0, 1]$ avoids extreme scaling from SNR values, improving conditioning and optimizer robustness.
>
>
> > the statement (Line 686) about the denoiser is incorrect, it states that "this estimator is mode-averaging, and may fall between mixture modes". Although its expression might give this deceptive impression, the authors disregard the fact that the weights of the components depend on $x_t$
>
>
> For any **fixed** \(x_t\), the posterior mean $m_{0|t}(x_t) = \sum_{k} r_k(x_t)\, m_k$ is a convex combination of the component-wise posterior means $m_k$, since
> $r_k(x_t) \ge 0$ and $\sum_k r_k(x_t) = 1$. Thus, whenever several components have non-negligible responsibilities for the same $x_t$, the posterior mean can indeed lie between those modes—this is the standard “mode-averaging’’ behavior of mixture-model posteriors. The fact that the responsibilities $r_k(x_t)$ depend on $x_t$ only determines **which** modes are averaged for that particular observation; it does not change the convex-combination property. To avoid ambiguity, we will revise the sentence to: “For a fixed $x_t$, the posterior mean is a responsibility-weighted average of the component-wise posterior means, and can fall between mixture modes when the conditional posterior is multimodal.”
>
> ## Reply to Weakness 3 - Typos/mistakes
>
> > - Line 230: it is not the correct expression of the $Cov(X_0\mid X_t=x_t)$, it must be $Cov(X_0\mid X_t=x_t)=\sigma_t/\alpha_t \nabla m_{0\mid t}$; see equation (9) in [3]
>
> Thanks to the reviewer for pointing out this typo.  The correct expression adapted to our notation is $Cov(X_0\mid X_t=x_t)=\sigma_t^2/\alpha_t \nabla m_{0\mid t}$, We have revised the expression in the paper accordingly.
>
> > - k is introduced in Equation 12, but it is not clear what it refers to
>
> In practice, we introduce a tunable parameter $k$ that adjusts the relative influence between the denoising estimate $m_{0 \mid t}$ and the measurement $y$.   See the above response as well.  We have added this explanation in the revised paper.
>
> > - Line 216-218: deforms what is being said in the [3] [see Proposition 1 in [3]], this actually the definition of the conditional covariance, and has nothing to do with "faithfully reflect the covariance of x0 | xt"
>
> Thank you, this was our intended statement and we revised it accordingly.

---

> > ### Author Response · Authors · 2025-11-21
> >
> > # Part 3
> >
> > ## Reply to Question 1
> >
> > > Can the authors provide clarification on Figure 3, namely the inverse problem setup? it seems the samples are generated from the prior distribution.
> >
> > This is a comparison between DPS, LMAPS and DAPS to do posterior sampling, not for solving inverse problems. Here we assume the posterior distribution is a Gaussian mixture, See Appendix A, then $\mathbb{E}[x_0 \mid x_{t_k}, y]$, $p(x_0 \mid x_{t_k}, y)$ and $\arg \max p(x_0 \mid x_{t_k}, y)$ have analytical solutions, and we can apply algorithms in Figure 2 to do DPS, LMAPS and DAPS. The goal of Figure 3 is to demonstrate that LMAPS is more likely to generate samples in high-density regions. We have added an explicit reference to App A accompanying Fig 3.
> >
> > ## Reply to Question 2
> >
> > > Can the authors provide clarification on the claim in Lines 302–305 about using $K$ gradient steps despite a closed-form solution? If the operator admits an SVD (e.g., inpainting), the advantage of iterative updates is unclear.
> >
> > We have added a direct comparison between solving $x_0^*$ using gradient-based optimization and the closed-form solution for SR $8\times$ and Inpainting (Box) in Table 8 and Appendix E.3. The results demonstrate that the performance of the two approaches is highly comparable. In particular, the PSNR difference is consistently below 0.5 dB, and the LPIPS difference is less than 0.015 across both FFHQ and ImageNet for these two linear inverse problems.
> >
> > These findings indicate that the closed-form solution achieves nearly identical reconstruction quality to gradient-based optimization while being more efficient. Therefore, when the forward operator admits an SVD, we recommend using the closed-form solution in practice.

---

### Official Review · Reviewer_Hjxk · 2025-11-04

**Soundness:** 2
**Presentation:** 2
**Contribution:** 2
**Rating:** 6
**Confidence:** 4

**Summary:**

This paper presents LMAPS, a new inference algorithm for solving inverse problems with pre-trained diffusion model. LMAPS operates by iteratively solving a local reformulated MAP problem. Empirically, LMAPS achieves strong recovery performance across 10 image restoration tasks and 3 scientific inverse problems.

**Strengths:**

- The objective reformulation is simple but effective.
- High recovery accuracy across a comprehensive list of benchmarks.
- Hyperparameters for each experiments are provided and their effects on the results are studied in the ablations.

**Weaknesses:**

- The introduction states “posterior sampling is not fully aligned with the objectives of inverse problem solving,” attributing single-GT evaluation to that philosophy. However, the fact that existing inverse problem benchmarks in ML community assume a single ground truth is mostly due to convenience. Setting up a single ground posterior or proper probabilistic tests is much more challenging than a single ground truth. That said, posterior sampling aspect remains important as it can capture the multi-modality, provide credible intervals, and uncertainty calibration, which are important in real-world decision making. Please soften this claim and acknowledge evaluation convenience vs. task desiderata.
- The claimed contribution of developing gradient approximation strategy for non-differentiable operators is overstated and under-specified.
	- Using a differentiable function to approximate the non-differentiable operator is a long-standing general idea. Even for the specific JPEG example used in this paper, there is already a line of differentiable JPEG works [1]. Claiming broad novelty here is problematic.
	- Surrogate choice is under-specified and trivialized by the JPEG example. The authors propose the existence of a differentiable surrogate and then instantiates it with the identity map for JPEG. This example is helpful as a baseline but does not address the central problem: how to construct a good surrogate precisely and systematically for a general forward model?
	- I would suggest remove the broad claim about general non-differentiable operators and only keep the JPEG example as it does not seem the focus of this work.
- In Sec. 4.3, the authors assume that when $H(x_{0})$ and $H^\prime(x_{0})$ are close, their gradients are also close. But this is not true without specifying the space of functions. Value closeness does not imply gradient closeness.
- The computation complexity scales with diffusion steps x gradient steps, where each inner step requires at least one forward model evaluation and one gradient call. This means the typical settings use around 200x200 forward operator evaluations. For problems where the forward model call is computationally expensive, LMAPS can be computationally expensive to run.
- Line 232, $\leq$ should be $\geq$?


[1]: Reich, Christoph, et al. "Differentiable jpeg: The devil is in the details." _Proceedings of the IEEE/CVF Winter Conference on Applications of Computer Vision_. 2024.

**Questions:**

- If the authors insist on maintaining the novelty claim regarding non-differentiable problems, please provide concrete and reproducible steps for constructing a good surrogate. Rather than the informal statement that “there exists a surrogate that is sufficiently close,” the paper should explicitly define:
	- the function space from which the surrogate is drawn,
	- the closeness metric used to quantify similarity between the surrogate and the true forward operator,
	- the computational procedure or algorithm for obtaining such a surrogate in general settings.
	- In particular, please address cases where the forward model does not admit a closed-form expression (e.g., physical processes governed by PDEs or other simulators), and explain how your approach can handle these scenarios in practice.
- How are the hyperparameters selected for each problem?

---

> ### Author Response · Authors · 2025-11-21
>
> ## Response to Weakness 1
> > The introduction states “posterior sampling is not fully aligned with the objectives of inverse problem solving,” attributing single-GT evaluation to that philosophy. However, the fact that existing inverse problem benchmarks in ML community assume a single ground truth is mostly due to convenience. Setting up a single ground posterior or proper probabilistic tests is much more challenging than a single ground truth. That said, posterior sampling aspect remains important as it can capture the multi-modality, provide credible intervals, and uncertainty calibration, which are important in real-world decision making. Please soften this claim and acknowledge evaluation convenience vs. task desiderata.
>
> We thank the reviewer for this insightful comment. We agree that posterior sampling is essential in Bayesian inverse problems for capturing multi-modality and providing calibrated uncertainty. Our intention was not to diminish its importance, but to highlight that MAP estimation has also been a long-standing and widely used objective in classical inverse problems. Accordingly, we have softened the phrasing in the introduction to reflect this balance and to clarify that MAP and posterior sampling serve complementary roles.
>
> ## Response to Weakness 2, 3 and Question 1
> > The claimed contribution of developing gradient approximation strategy for non-differentiable operators is overstated and under-specified.
>
> > In Sec. 4.3, the authors assume that when $H(x_0)$ and $H'(x_0)$ are close, their gradients are also close. But this is not true without specifying the space of functions. Value closeness does not imply gradient closeness.
>
> We removed the broad claim about general non-differentiable operators and only kept the JPEG example.
>
> ## Response to Weakness 4
> > The computation complexity scales with diffusion steps x gradient steps, where each inner step requires at least one forward model evaluation and one gradient call. This means the typical settings use around 200x200 forward operator evaluations. For problems where the forward model call is computationally expensive, LMAPS can be computationally expensive to run.
>
> In cases where the forward model call is computationally expensive, LMAPS can also achieve competitive performance with fewer forward operator evaluations, e.g., reducing the inner gradient steps. For example,  the forward model call of nonlinear deblurring is computationally expensive. For nonlinear deblurring on FFHQ dataset, reducing the diffusion step to 100 and inner optimization step to 20, LMAPS can still achieve PSNR of 27.57, SSIM of 0.814 and LPIPS of 0.200. We added additional results on solving nonlinear deblurring with 100 diffusion steps and 20 inner optimization steps in Table 9.
>
> Besides, there are some potential directions to further improve the speed under the LMAPS framework: (1) it's possible to develop other faster optimization solvers for local MAP; (2) appropriate optimization steps schedules and early stopping techniques might be helpful to improve the sampling speed.
>
>
>
> ## Response to Weakness 5
> > Line 232, $\leq$ should be $\geq$?
>
> For a Gaussian prior $x_0 \sim \mathcal{N}(\mu_0, \Sigma_0)$, the exact posterior covariance under the forward noising process $x_t = \alpha_t x_0 + \sigma_t \epsilon$, $\epsilon \sim \mathcal{N}(0,\mathbb{I})$ is
>
> $$\Sigma_{0 \mid t}
>     = \bigl(\Sigma_0^{-1} + \tfrac{\alpha_t^2}{\sigma_t^2} \mathbb{I}\bigr)^{-1}
>     = \frac{\sigma_t^2}{\alpha_t^2} \mathbb{I}
>       + \mathcal{O} \left( \bigl(\tfrac{\sigma_t^2}{\alpha_t^2}\bigr)^2 \right)
>     \preceq \frac{\sigma_t^2}{\alpha_t^2} \mathbb{I},$$
> We provide detailed proof in the Appendix B.
>
>
> ## Response to Question 2
> > How are the hyperparameters selected for each problem?
>
> For each inverse problem, the hyperparameters $k_1$ and $k_2$ are selected based on task-specific tuning, but there are some experiences and rules for choosing the hyperparameters.
>
> **$k$.** In practice, we choose `k1` in `[0, 1]`; values slightly larger than the measurement noise standard deviation $\sigma_y$ work well empirically across tasks.
>
> **$\eta$.** $\eta$ is the learning rate, we choose $\eta$ in [0.005, 1], for scientific inverse problems.
>
> **$k_2$.** We select $k_2$ starting from $k_2=1/ \eta$, and tuning it to find the best performance. For image restoration like SR $4\times$, Inpaint (Box), Inpainting (Random), Gaussian Deblurring, Motion Deblurring, HDR, JPEG restoration and Quantization. we all set $k_2=1 /\eta$.

---

### Author Response · Authors · 2025-12-03
**General response**

We thank the reviewers for their detailed feedback and for recognizing the effectiveness of LMAPS in achieving state-of-the-art performance. We have uploaded a revised manuscript with the following key updates:

**MAP vs. Posterior Sampling (Addressed R1, R2)**. We revised the Introduction to remove the claim that posterior sampling is "misaligned" with inverse problems. We now explicitly acknowledge that posterior sampling is essential for uncertainty quantification, while clarifying that our focus is on point estimation (MAP), which remains a standard objective for restoration benchmarks.

**Non-Differentiable Operators (Addressed R1, R4)**. We have removed the broad claims regarding general non-differentiable operators. As suggested, we now present the JPEG restoration as a specific case study rather than a general theoretical contribution for all non-differentiable functions.

**Theoretical Analysis of LMAPS biased towards highly-likely regions (Addressed R3)**. To provide a concrete theoretical grounding, we added Appendix A, analyzing a Gaussian Mixture prior. We prove that LMAPS consistently concentrates on dominant posterior modes while avoiding low-density regions, confirming its theoretical distinction from standard posterior sampling (DPS).

**Computational Efficiency (Addressed R1)**. We added experiments on nonlinear deblurring (Table 9) using fewer steps (100 diffusion, 20 inner optimization). The results show LMAPS maintains competitive performance even with significantly reduced computational cost.

**Additional Baselines and Validations (Addressed R3)**. We added comparisons to MGDM, MMPS, DMPlug and MAP-GA in **Table 1**. LMAPS continues to outperform these recent methods in the majority of tasks (e.g., 17 out of 18 cases vs. MAP-GA/DMPlug). We added **Table 8** comparing our gradient-based solver to the closed-form solution for linear problems. The results are nearly identical, validating the accuracy of our optimization approach.

---

### Meta-Review · Area_Chair_JyNf · 2026-01-07

**Summary:**

This paper proposes a framework for solving inverse problems with pre-trained diffusion models. As noted by reviewers Hjxk and Bng9, the claims of the contribution appear overstated, and the manuscript contains several technical inaccuracies and confusing statements. In addition, reviewer  8LcP and Bng9 mentioned that the comparison with and the discussions of existing works are either insufficient or inaccurate. Based on these limitations, I agree with the reviewers and recommend rejection.

**Reviewer Concerns:**

While minor comments such as the typos noted by reviewer 8LcP have been addressed, the concerns stated in the summary are still outstanding.

**Reviewer Scores:**

There is no clear evidence to suggest that any reviewer would have changed their score after a full discussion.

---

### Decision · Program_Chairs · 2026-01-26

Reject